# Linear Time Sinkhorn Divergences using Positive Features

**Meyer Scetbon**
CREST, ENSAE,
Institut Polytechnique de Paris,
meyer.scetbon@ensae.fr

**Marco Cuturi**
Google Brain,
CREST, ENSAE,
cuturi@google.com

## Abstract

Although Sinkhorn divergences are now routinely used in data sciences to compare probability distributions, the computational effort required to compute them remains expensive, growing in general quadratically in the size $n$ of the support of these distributions. Indeed, solving optimal transport (OT) with an entropic regularization requires computing a $n \times n$ kernel matrix (the neg-exponential of a $n \times n$ pairwise ground cost matrix) that is repeatedly applied to a vector. We propose to use instead ground costs of the form $c(x, y) = -\log\langle\varphi(x), \varphi(y)\rangle$ where $\varphi$ is a map from the ground space onto the positive orthant $\mathbb{R}^r_+$, with $r \ll n$. This choice yields, equivalently, a kernel $k(x, y) = \langle\varphi(x), \varphi(y)\rangle$, and ensures that the cost of Sinkhorn iterations scales as $O(nr)$. We show that usual cost functions can be approximated using this form. Additionally, we take advantage of the fact that our approach yields approximation that remain fully differentiable with respect to input distributions, as opposed to previously proposed adaptive low-rank approximations of the kernel matrix, to train a faster variant of OT-GAN [49].

## 1 Introduction

Optimal transport (OT) theory [56] plays an increasingly important role in machine learning to compare probability distributions, notably point clouds, discrete measures or histograms [43]. As a result, OT is now often used in graphics [11, 44, 45], neuroimaging [33], to align word embeddings [4, 1, 30], reconstruct cell trajectories [32, 50, 58], domain adaptation [14, 15] or estimation of generative models [5, 49, 26]. Yet, in their original form, as proposed by Kantorovich [34], OT distances are not a natural fit for applied problems: they minimize a network flow problem, with a supercubic complexity $(n^3 \log n)$ [55] that results in an output that is *not* differentiable with respect to the measures' locations or weights [10, §5]; they suffer from the curse of dimensionality [18, 22] and are therefore likely to be meaningless when used on samples from high-dimensional densities.

Because of these statistical and computational hurdles, all of the works quoted above do rely on some form of regularization to smooth the OT problem, and some more specific uses of an entropic penalty, to recover so called Sinkhorn divergences [16]. These divergences are cheaper to compute than regular OT [12, 24], smooth and programmatically differentiable in their inputs [11, 32], and have a better sample complexity [28] while still defining convex and definite pseudometrics [21]. While Sinkhorn divergences do lower OT costs from supercubic down to an embarassingly parallel quadratic cost, using them to compare measures that have more than a few tens of thousands of points in forward mode (less obviously if backward execution is also needed) remains a challenge.

**Entropic regularization: starting from ground costs.** The definition of Sinkhorn divergences usually starts from that of the ground cost on observations. That cost is often chosen by default to be a $q$-norm between vectors, or a shortest-path distance on a graph when considering geometric domains [29, 52, 53, 33]. Given two measures supported respectively on $n$ and $m$ points, regularized

OT instantiates first a $n \times m$ pairwise matrix of costs $C$, to solve a linear program penalized by the coupling's entropy. This can be rewritten as a Kullback-Leibler minimization:

$$\min_{\text{couplings } \mathbf{P}} \langle \mathbf{C}, \mathbf{P} \rangle - \varepsilon H(\mathbf{P}) = \varepsilon \min_{\text{couplings } \mathbf{P}} \text{KL}(\mathbf{P} \| \mathbf{K}), \qquad (1)$$

where matrix $K$ appearing in Eq. (1) is defined as $\mathbf{K} := \exp(-\mathbf{C}/\varepsilon)$, the elementiwe neg-exponential of a rescaled cost $\mathbf{C}$.As described in more detail in §2, this problem can then be solved using Sinkhorn's algorithm, which only requires applying repeatedly kernel $\mathbf{K}$ to vectors. While faster optimization schemes to compute regularized OT have been been investigated [2, 19, 37], the Sinkhorn algorithm remains, because of its robustness and simplicity of its parallelism, the workhorse of choice to solve entropic OT. Since Sinkhorn's algorithm cost is driven by the cost of applying $\mathbf{K}$ to a vector, speeding up that evaluation is the most impactful way to speedup Sinkhorn's algorithm. This is the case when using separable costs on grids (applying $\mathbf{K}$ boils down to carrying out a convolution at cost $(n^{1+1/d})$ [43, Remark 4.17]) or when using shortest path metrics on graph in which case applying $\mathbf{K}$ can be approximated using a heat-kernel [54]. While it is tempting to use low-rank matrix factorization, using them within Sinkhorn iterations requires that the application of the approximated kernel guarantees the positiveness of the output. As shown by [3] this can only be guaranteed, when using the Nyström method, when regularization is high and tolerance very low.

**Starting instead from the Kernel.** Because regularized OT can be carried out using only the definition of a kernel $\mathbf{K}$, we focus instead on kernels $\mathbf{K}$ that are guaranteed to have positive entries by design. Indeed, rather than choosing a cost to define a kernel next, we consider instead ground costs of the form $c(x, y) = -\varepsilon \log \langle \varphi(x), \varphi(y) \rangle$ where $\varphi$ is a map from the ground space onto the positive orthant in $\mathbb{R}^r$. This choice ensures that both the Sinkhorn algorithm itself (which can approximate optimal primal and dual variables for the OT problem) and the evaluation of Sinkhorn divergences can be computed exactly with an effort scaling linearly in $r$ and in the number of points, opening new perspectives to apply OT at scale.

**Our contributions** are two fold: *(i)* We show that kernels built from positive features can be used to approximate some usual cost functions including the square Euclidean distance using random expansions. *(ii)* We illustrate the versatility of our approach by extending previously proposed OT-GAN approaches [49, 28], that focused on learning adversarially cost functions $c_\theta$ and incurred therefore a quadratic cost, to a new approach that learns instead adversarially a kernel $k_\theta$ induced from a positive feature map $\varphi_\theta$. We leverage here the fact that our approach is fully differentiable in the feature map to train a GAN at scale, with linear time iterations.

**Notations.** Let $\mathcal{X}$ be a compact space endowed with a cost function $c : \mathcal{X} \times \mathcal{X} \to \mathbb{R}$ and denote $D = \sup_{(x,y) \in \mathcal{X} \times \mathcal{X}} \|(x, y)\|_2$. We denote $\mathcal{P}(\mathcal{X})$ the set of probability measures on $\mathcal{X}$. For all $n \geq 1$, we denote by $\Delta_n$ all vectors in $\mathbb{R}_+^n$ with positive entries and summing to 1. We denote $f \in \mathcal{O}(g)$ if $f \leq Cg$ for a universal constant $C$ and $f \in \Omega(g)$ if $g \leq Qf$ for a universal constant $Q$.

## 2 Regularized Optimal Transport

**Sinkhorn Divergence.** Let $\mu = \sum_{i=1}^n a_i \delta_{x_i}$ and $\nu = \sum_{j=1}^m b_j \delta_{y_j}$ be two discrete probability measures. The Sinkhorn divergence [48, 27, 49] between $\mu$ and $\nu$ is, given a constant $\varepsilon > 0$, equal to

$$\overline{W}_{\varepsilon,c}(\mu, \nu) := W_{\varepsilon,c}(\mu, \nu) - \frac{1}{2} \left( W_{\varepsilon,c}(\mu, \mu) + W_{\varepsilon,c}(\nu, \nu) \right), \text{ where} \qquad (2)$$

$$W_{\varepsilon,c}(\mu, \nu) := \min_{\substack{P \in \mathbb{R}_+^{n \times m} \\ P\mathbf{1}_m = a, P^T \mathbf{1}_n = b}} \langle P, C \rangle - \varepsilon H(P) + \varepsilon. \qquad (3)$$

Here $\mathbf{C} := [c(x_i, y_j)]_{ij}$ and $H$ is the Shannon entropy, $H(\mathbf{P}) := -\sum_{ij} P_{ij}(\log P_{ij} - 1)$. Because computing and differentiating $\overline{W}_{\varepsilon,c}$ is equivalent to doing so for three evaluations of $W_{\varepsilon,c}$ (neglecting the third term in the case where only $\mu$ is a variable) [43, §4], we focus on $W_{\varepsilon,c}$ in what follows.

**Primal Formulation.** Problem (3) is $\varepsilon$-strongly convex and admits therefore a unique solution $\mathbf{P}^\star$ which, writing first order conditions for problem (3), admits the following factorization:

$$\exists u^\star \in \mathbb{R}_+^n, v^\star \in \mathbb{R}_+^m \text{ s.t. } \mathbf{P}^\star = \text{diag}(u^\star) \mathbf{K} \text{diag}(v^\star), \text{ where } \mathbf{K} := \exp(-\mathbf{C}/\varepsilon). \qquad (4)$$

These *scalings* $u^\star, v^\star$ can be computed using Sinkhorn's algorithm, which consists in initializing $u$ to any arbitrary positive vector in $\mathbb{R}^m$, to apply then fixed point iteration described in Alg. 1.

These two iterations require together $2nm$ operations if $\mathbf{K}$ is stored as a matrix and applied directly. The number of Sinkhorn iterations needed to converge to a precision $\delta$ (monitored by the difference between the column-sum of $\text{diag}(u)\mathbf{K}\text{diag}(v)$ and $b$) is controlled by the scale of elements in $C$ relative to $\varepsilon$ [23]. That convergence deteriorates with smaller $\varepsilon$, as studied in more detail by [57, 20].

---
**Algorithm 1** Sinkhorn

**Inputs:** $\mathbf{K}, a, b, \delta, u$ **repeat**
  | $v \leftarrow b/\mathbf{K}^T u, \ u \leftarrow a/\mathbf{K}v$
**until** $\|v \circ \mathbf{K}^T u - b\|_1 < \delta$;
**Result:** $u, v$

---

**Dual Formulation.** The dual of (3) plays an important role in our analysis [43, §4.4]:

$$W_{\varepsilon,c}(\mu,\nu) = \max_{\alpha\in\mathbb{R}^n,\beta\in\mathbb{R}^m} a^T\alpha + b^T\beta - \varepsilon(e^{\alpha/\varepsilon})^T\mathbf{K}e^{\beta/\varepsilon} + \varepsilon = \varepsilon\left(a^T\log u^\star + b^T\log v^\star\right) \quad (5)$$

where we have introduced, next to its definition, its evaluation using optimal scalings $u^\star$ and $v^\star$ described above. This equality comes from that fact that *(i)* one can show that $\alpha^\star := \varepsilon\log u^\star$, $\beta^\star := \varepsilon\log v^\star$, *(ii)* the term $(e^{\alpha/\varepsilon})^T\mathbf{K}e^{\beta/\varepsilon} = u^T\mathbf{K}v$ is equal to 1, whenever the Sinkhorn loop has been applied even just once, since these sums describe the sum of a coupling (a probability distribution of size $n \times m$). As a result, given the outputs $u, v$ of Alg. 1 we estimate (3) using

$$\widehat{W}_{\varepsilon,c}(\mu,\nu) = \varepsilon\left(a^T\log u + b^T\log v\right). \quad (6)$$

Approximating $W_{\varepsilon,c}(\mu,\nu)$ can be therefore carried using exclusively calls to the Sinkhorn algorithm, which requires instantiating kernel $\mathbf{K}$, in addition to computing inner product between vectors, which can be computed in $\mathcal{O}(n + m)$ algebraic operations; the instantiation of $\mathbf{C}$ is never needed, as long as $\mathbf{K}$ is given. Using this dual formulation(3) we can now focus on kernels that can be evaluated with a linear cost to achieve linear time Sinkhorn divergences.

## 3 Linear Sinkhorn with Positive Features

The usual flow in transport dictates to choose a cost first $c(x,y)$ to define a kernel $k(x,y) := \exp(-c(x,y)/\varepsilon)$ next, and adjust the temperature $\varepsilon$ depending on the level of regularization that is adequate for the task. We propose in this work to do exactly the opposite, by choosing instead parameterized feature maps $\varphi_\theta : \mathcal{X} \mapsto (\mathbb{R}_+^*)^r$ which associate to any point in $\mathcal{X}$ a vector in the positive orthant. With such maps, we can therefore build the corresponding positive-definite kernel $k_\theta$ as $k_\theta(x,y) := \varphi_\theta(x)^T\varphi_\theta(y)$ which is a positive function. Therefore as a by-product and by positivity of the feature map, we can define for all $(x,y) \in \mathcal{X} \times \mathcal{X}$ the following cost function

$$c_\theta(x,y) := -\varepsilon\log\varphi_\theta(x)^T\varphi_\theta(y). \quad (7)$$

**Remark 1** (Transport on the Positive Sphere.)**.** *Defining a cost as the log of a dot-product as described in (7) has already played a role in the recent OT literature. In [42], the author defines a cost $c$ on the sphere $\mathbb{S}^d$, as $c(x,y) = -\log x^T y$, if $x^T y > 0$, and $\infty$ otherwise. The cost is therefore finite whenever two normal vectors share the same halfspace, and infinite otherwise. When restricted to the the positive sphere, the kernel associated to this cost is the linear kernel. See App. C for an illustration.*

More generally, the above procedure allows us to build cost functions on any cartesian product spaces $\mathcal{X} \times \mathcal{Y}$ by defining $c_{\theta,\gamma}(x,y) := -\varepsilon\log\varphi_\theta(x)^T\psi_\gamma(y)$ where $\psi_\gamma : \mathcal{Y} \mapsto (\mathbb{R}_+^*)^r$ is a parametrized function which associates to any point $\mathcal{Y}$ also a vector in the same positive orthant as the image space of $\varphi_\theta$ but this is out of the scope of this paper.

### 3.1 Achieving linear time Sinkhorn iterations with Positive Features

Choosing a cost function $c_\theta$ as in (7) greatly simplifies computations, by design, since one has, writing for the matrices of features for two set of points $x_1, \ldots, x_n$ and $y_1, \ldots, y_m$

$$\boldsymbol{\xi} := [\varphi_\theta(x_1), \ldots, \varphi_\theta(x_n)] \in (\mathbb{R}_+^*)^{r\times n}, \qquad \boldsymbol{\zeta} := [\varphi_\theta(y_1), \ldots, \varphi_\theta(y_m)] \in (\mathbb{R}_+^*)^{r\times m},$$

that the resulting sample kernel matrix $\mathbf{K}_\theta$ corresponding to the cost $c_\theta$ is $\mathbf{K}_\theta = \left[e^{-c_\theta(x_i,y_j)/\varepsilon}\right]_{i,j} = \boldsymbol{\xi}^T\boldsymbol{\zeta}$. Moreover thanks to the positivity of the entries of the kernel matrix $\mathbf{K}_\theta$ there is no duality gap and we obtain that

$$W_{\varepsilon,c_\theta}(\mu,\nu) = \max_{\alpha\in\mathbb{R}^n,\beta\in\mathbb{R}^m} a^T\alpha + b^T\beta - \varepsilon(\boldsymbol{\xi}e^{\alpha/\varepsilon})^T\boldsymbol{\zeta}e^{\beta/\varepsilon} + \varepsilon. \quad (8)$$

Therefore the Sinkhorn iterations in Alg. 1 can be carried out in exactly $r(n+m)$ operations. The main question remains on how to choose the mapping $\varphi_\theta$. In the following, we show that, for some well chosen mappings $\varphi_\theta$, we can approximate the ROT distance for some classical costs in linear time.

## 3.2 Approximation properties of Positive Features

Let $\mathcal{U}$ be a metric space and $\rho$ a probability measure on $\mathcal{U}$. We consider kernels on $\mathcal{X}$ of the form:

$$\text{for } (x,y) \in \mathcal{X}^2,\ k(x,y) = \int_{u \in \mathcal{U}} \varphi(x,u)^T \varphi(y,u) d\rho(u). \tag{9}$$

Here $\varphi : \mathcal{X} \times \mathcal{U} \to (\mathbb{R}_+^*)^p$ is such that for all $x \in \mathcal{X}$, $u \in \mathcal{U} \to \|\varphi(x,u)\|_2$ is square integrable (for the measure $d\rho$). Given such kernel and a regularization $\varepsilon$ we define the cost function $c(x,y) := -\varepsilon \log(k(x,y))$. In fact, we will see in the following that for some usual cost functions $\tilde{c}$, e.g. the square Euclidean cost, the Gibbs kernel associated $\tilde{k}(x,y) = \exp(-\varepsilon^{-1}\tilde{c}(x,y))$ admits a decomposition of the form Eq.(9). To obtain a finite-dimensional representation, one can approximate the integral with a weighted finite sum. Let $r \geq 1$ and $\theta := (u_1, ..., u_r) \in \mathcal{U}^r$ from which we define the following positive feature map

$$\varphi_\theta(x) := \frac{1}{\sqrt{r}} \left( \varphi(x,u_1), ..., \varphi(x,u_r) \right) \in \mathbb{R}^{p \times r}$$

and a new kernel as $k_\theta(x,y) := \langle \varphi_\theta(x), \varphi_\theta(y) \rangle$. When the $(u_i)_{1 \leq i \leq r}$ are sampled independently from $\rho$, $k_\theta$ may approximates the kernel $k$ arbitrary well if the number of random features $r$ is sufficiently large. For that purpose let us now introduce some assumptions on the kernel $k$.

**Assumption 1.** *There exists a constant $\psi > 0$ such that for all $x,y \in \mathcal{X}$:*

$$|\varphi(x,u)^T \varphi(y,u)/k(x,y)| \leq \psi \tag{10}$$

**Assumption 2.** *There exists a $\kappa > 0$ such that for ally $x,y \in \mathcal{X}$, $k(x,y) \geq \kappa > 0$ and $\varphi$ is differentiable there exists $V > 0$ such that:*

$$\sup_{x \in \mathcal{X}} \mathbf{E}_\rho \left( \|\nabla_x \varphi(x,u)\|^2 \right) \leq V \tag{11}$$

We can now present our main result on our proposed approximation scheme of $W_{\varepsilon,c}$ which is obtained in linear time with high probability. See Appendix A.1 for the proof.

**Theorem 3.1.** *Let $\delta > 0$ and $r \geq 1$. Then the Sinkhorn Alg. 1 with inputs $\mathbf{K}_\theta$, $a$ and $b$ outputs $(u_\theta, v_\theta)$ such that $|W_{\varepsilon,c_\theta} - \widehat{W}_{\varepsilon,c_\theta}| \leq \frac{\delta}{2}$ in $\mathcal{O}\left( \frac{n\varepsilon r}{\delta} \left[ Q_\theta - \log \min_{i,j}(a_i, b_j) \right]^2 \right)$ algebric operations where $Q_\theta = -\log \min_{i,j} k_\theta(x_i, y_j)$. Moreover if Assumptions 1 and 2 hold then for $\tau > 0$,*

$$r \in \Omega \left( \frac{\psi^2}{\delta^2} \left[ \min \left( d\varepsilon^{-1}\|\mathbf{C}\|_\infty^2 + d\log\left(\frac{\psi V D}{\tau\delta}\right), \log\left(\frac{n}{\tau}\right) \right) \right] \right) \tag{12}$$

*and $u_1, ..., u_r$ drawn independently from $\rho$, with a probability $1 - \tau$, $Q_\theta \leq \varepsilon^{-1}\|\mathbf{C}\|_\infty^2 + \log\left(2 + \delta\varepsilon^{-1}\right)$ and it holds*

$$|W_{\varepsilon,c} - \widehat{W}_{\varepsilon,c_\theta}| \leq \delta \tag{13}$$

Therefore with a probability $1 - \tau$, Sinkhorn Alg. 1 with inputs $\mathbf{K}_\theta$, $a$ and $b$ output a $\delta$-approximation of the ROT distance in $\tilde{\mathcal{O}}\left( \frac{n}{\varepsilon\delta^3}\|\mathbf{C}\|_\infty^4 \psi^2 \right)$ algebraic operation where the notation $\tilde{\mathcal{O}}(.)$ omits polylog-arithmic factors depending on $R, D, \varepsilon, n$ and $\delta$.

It worth noting that for every $r \geq 1$ and $\theta$, Sinkhorn Alg. 1 using kernel matrix $\mathbf{K}_\theta$ will converge towards an approximate solution of the ROT problem associated with the cost function $c_\theta$ in linear time thanks to the positivity of the feature maps used. Moreover, to ensure with high probability that the solution obtained approximate an optimal solution for the ROT problem associated with the cost function $c$, we need, if the features are chosen randomly, to ensure a minimum number of them. In constrast such result is not possible in [3]. Indeed in their works, the number of random features $r$ cannot be chosen arbitrarily as they need to ensure the positiveness of the all the coefficients of the approximated kernel matrix obtained by the Nyström algorithm of [40] to run the Sinkhorn iterations and therefore need a very high precision which requires a certain number of random features $r$.

**Remark 2** (Acceleration.). *It is worth noting that our method can also be applied in combination with the accelerated version of the Sinkhorn algorithm proposed in [31]. Indeed for $\tau > 0$, applying our approximation scheme to their algorithm leads with a probability $1 - \tau$ to a $\delta/2$-approximation of $W_{\varepsilon,c}$ in $\mathcal{O}\left(\frac{nr}{\sqrt{\delta}}[\sqrt{\varepsilon^{-1}}A_\theta]\right)$ algebraic operations where $A_\theta = \inf\limits_{(\alpha,\beta)\in\Theta_\theta} \|(\alpha,\beta)\|_2$, $\Theta_\theta$ is the set of optimal dual solutions of (8) and $r$ satisfying Eq.(12). See the full statement and the proof in Appendix A.2.*

The number of random features prescribed in Theorem 3.1 ensures with high probability that $\widehat{W}_{\varepsilon,c_\theta}$ approximates $W_{\varepsilon,c}$ well when $u_1, \ldots, u_r$ are drawn independently from $\rho$. Indeed, to control the error due to the approximation made through the Sinkhorn iterations, we need to control the error of the approximation of $\mathbf{K}$ by $\mathbf{K}_\theta$ relatively to $\mathbf{K}$. In the next proposition we show with high probability that for all $(x,y) \in \mathcal{X} \times \mathcal{X}$,

$$(1 - \delta)k(x,y) \leq k_\theta(x,y) \leq (1 + \delta)k(x,y) \tag{14}$$

for an arbitrary $\delta > 0$ as soon as the number of random features $r$ is large enough. See Appendix A.3 for the proof.

**Proposition 3.1.** *Let $\mathcal{X} \subset \mathbb{R}^d$ compact, $n \geq 1$, $\mathbf{X} = \{x_1, ..., x_n\}$ and $\mathbf{Y} = \{y_1, ..., y_n\}$ such that $\mathbf{X}, \mathbf{Y} \subset \mathcal{X}$, $\delta > 0$. If $u_1, ..., u_r$ are drawn independently from $\rho$ then under Assumption 1 we have*

$$\mathbb{P}\left(\sup_{(x,y)\in\mathbf{X}\times\mathbf{Y}} \left|\frac{k_\theta(x,y)}{k(x,y)} - 1\right| \geq \delta\right) \leq 2n^2 \exp\left(-\frac{r\delta^2}{2\psi^2}\right)$$

*Moreover if in addition Assumption 2 holds then we have*

$$\mathbb{P}\left(\sup_{(x,y)\in\mathcal{X}\times\mathcal{X}} \left|\frac{k_\theta(x,y)}{k(x,y)} - 1\right| \geq \delta\right) \leq \frac{(\kappa^{-1}D)^2 C_{\psi,V,r}}{\delta^2} \exp\left(-\frac{r\delta^2}{2\psi^2(d+1)}\right)$$

*where $C_{\psi,V,r} = 2^9\psi(4 + \psi^2/r)V \sup\limits_{x\in\mathcal{X}} k(x,x)$ and $D = \sup\limits_{(x,y)\in\mathcal{X}\times\mathcal{X}} \|(x,y)\|_2$.*

**Remark 3** (Ratio Approximation.). *The uniform bound obtained here to control the ratio gives naturally a control of the form Eq.(14). In comparison, in [47], the authors obtain a uniform bound on their difference which leads with high probability to a uniform control of the form*

$$k(x,y) - \tau \leq k_\theta(x,y) \leq k(x,y) + \tau \tag{15}$$

*where $\tau$ is a decreasing function with respect to $r$ the number of random features required. To be able to recover Eq.(14) from the above control, one may consider the case when $\tau = \inf_{x,y\in\mathbf{X}\times\mathbf{Y}} k(x,y)\delta$ which can considerably increases the number of of random features $r$ needed to ensure the result with at least the same probability. For example if the kernel is the Gibbs kernel associated to a cost function c, then $\inf\limits_{x,y\in\mathbf{X}\times\mathbf{Y}} k(x,y) = \exp(-\|\mathbf{C}\|_\infty/\varepsilon)$. More details are left in Appendix A.3.*

In the following, we provides examples of some usual kernels $k$ that admits a decomposition of the form Eq.(9), satisfy Assumptions 1 and 2 and hence for which Theorem 3.1 can be applied.

**Arc-cosine Kernels.** Arc-cosine kernels have been considered in several works, starting notably from [51], [13] and [6]. The main idea behind arc-cosine kernels is that they can be written using positive maps for vectors $x, y$ in $\mathbb{R}^d$ and the signs (or higher exponent) of random projections $\mu = \mathcal{N}(0, I_d)$

$$k_s(x,y) = \int_{\mathbb{R}^d} \Theta_s(u^T x)\Theta_s(u^T y)d\mu(u)$$

where $\Theta_s(w) = \sqrt{2}\max(0, w)^s$ is a rectified polynomial function. In fact from these formulations, we build a perturbed version of $k_s$ which admits a decomposition of the form Eq.(9) that satisfies the required assumptions. See Appendix A.5 for the full statement and the proof.

**Gaussian kernel.** The Gaussian kernel is in fact an important example as it is both a very widely used kernel on its own and its cost function associated is the square Euclidean metric. A decomposition of the form (9) has been obtained in ([39]) for the Gaussian kernel but it does not satisfies the required assumptions. In the following lemma, we built a feature map of the Gaussian kernel that satisfies them. See Appendix A.4 for the proof.

**Lemma 1.** *Let $d \geq 1$, $\varepsilon > 0$ and $k$ be the kernel on $\mathbb{R}^d$ such that for all $x, y \in \mathbb{R}^d$, $k(x,y) = e^{-\|x-y\|_2^2/\varepsilon}$. Let $R > 0$, $q = \frac{R^2}{2\varepsilon d W_0(R^2/\varepsilon d)}$ where $W_0$ is the Lambert function, $\sigma^2 = q\varepsilon/4$, $\rho = \mathcal{N}\left(0, \sigma^2 Id\right)$ and let us define for all $x, u \in \mathbb{R}^d$ the following map*

$$\varphi(x, u) = (2q)^{d/4} \exp\left(-2\varepsilon^{-1}\|x-u\|_2^2\right) \exp\left(\frac{\varepsilon^{-1}\|u\|_2^2}{\frac{1}{2} + \varepsilon^{-1}R^2}\right)$$

*Then for any $x, y \in \mathbb{R}^d$ we have $k(x,y) = \int_{u \in \mathbb{R}^d} \varphi(x,u)\varphi(y,u)d\rho(u)$. Moreover if $x, y \in \mathcal{B}(0, R)$ and $u \in \mathbb{R}^d$ we have $k(x,y) \geq \exp(-4\varepsilon^{-1}R^2) > 0$,*

$$|\varphi(x,u)\varphi(y,u)/k(x,y)| \leq 2^{d/2+1}q^{d/2} \quad and \quad \sup_{x \in \mathcal{B}(0,R)} \mathbf{E}(\|\nabla_x \varphi\|_2^2) \leq 2^{d/2+3}q^{d/2}\left[(R/\varepsilon)^2 + \frac{q}{4\varepsilon}\right].$$

### 3.3 Constructive approach to Designing Positive Features: Differentiability

In this section we consider a constructive way of building feature map $\varphi_\theta$ which may be chosen arbitrary, or learned accordingly to an objective defined as a function of the ROT distance, e.g. OT-GAN objectives [49, 25]. For that purpose, we want to be able to compute the gradient of $W_{\varepsilon,c_\theta}(\mu,\nu)$ with respect to the kernel $\mathbf{K}_\theta$, or more specifically with respect to the parameter $\theta$ and the locations of the input measures. In the next proposition we show that the ROT distance is differentiable with respect to the kernel matrix. See Appendix B for the proof.

**Proposition 3.2.** *Let $\epsilon > 0$, $(a,b) \in \Delta_n \times \Delta_m$ and let us also define for any $\mathbf{K} \in (\mathbb{R}_+^*)^{n \times m}$ with positive entries the following function:*

$$G(\mathbf{K}) := \sup_{(\alpha,\beta) \in \mathbb{R}^n \times \mathbb{R}^m} \langle \alpha, a \rangle + \langle \beta, a \rangle - \varepsilon(e^{\alpha/\varepsilon})^T \mathbf{K} e^{\beta/\varepsilon}. \tag{16}$$

*Then $G$ is differentiable on $(\mathbb{R}_+^*)^{n \times m}$ and its gradient is given by*

$$\nabla G(\mathbf{K}) = -\varepsilon e^{\alpha^*/\varepsilon}(e^{\beta^*/\varepsilon})^T \tag{17}$$

*where $(\alpha^*, \beta^*)$ are optimal solutions of Eq.(16).*

Note that when $c$ is the square euclidean metric, the differentiability of the above objective has been obtained in [17]. We can now provide the formula for the gradients of interest. For all $\mathbf{X} := [x_1, \ldots, x_n] \in \mathbb{R}^{d \times n}$, we denote $\mu(\mathbf{X}) = \sum_{i=1}^n a_i \delta_{x_i}$ and $W_{\varepsilon,c_\theta} = W_{\varepsilon,c_\theta}(\mu(\mathbf{X}), \nu)$. Assume that $\theta$ is a $M$-dimensional vector for simplicity and that $(x, \theta) \in \mathbb{R}^d \times \mathbb{R}^M \to \varphi_\theta(x) \in (\mathbb{R}_+^*)^r$ is a differentiable map. Then from proposition 3.2 and by applying the chain rule theorem, we obtain that

$$\nabla_\theta W_{\varepsilon,c_\theta} = -\varepsilon\left(\left(\frac{\partial \boldsymbol{\xi}}{\partial \theta}\right)^T u_\theta^\star(\boldsymbol{\zeta}v_\theta^\star)^T + \left(\frac{\partial \boldsymbol{\zeta}}{\partial \theta}\right)^T v_\theta^\star(\boldsymbol{\xi}u_\theta^\star)^T\right), \quad \nabla_X W_{\varepsilon,c_\theta} = -\varepsilon\left(\frac{\partial \boldsymbol{\xi}}{\partial X}\right)^T u_\theta^\star(\boldsymbol{\zeta}v_\theta^\star)^T$$

where $(u_\theta^*, v_\theta^*)$ are optimal solutions of (5) associated to the kernel matrix $\mathbf{K}_\theta$. Note that $\left(\frac{\partial \boldsymbol{\xi}}{\partial \theta}\right)^T$, $\left(\frac{\partial \boldsymbol{\zeta}}{\partial \theta}\right)^T$ and $\left(\frac{\partial \boldsymbol{\xi}}{\partial X}\right)^T$ can be evaluated using simple differentiation if $\varphi_\theta$ is a simple random feature, or, more generally, using automatic differentiation if $\varphi_\theta$ is the output of a neural network.

**Discussion.** Our proposed method defines a kernel matrix $\mathbf{K}_\theta$ and a parametrized ROT distance $W_{\varepsilon,c_\theta}$ which are differentiable with respect to the input measures and the parameter $\theta$. These proprieties are important and used in many applications, e.g. GANs. However such operations may not be allowed when using a data-dependent method to approximate the kernel matrix such as the Nyström method used in [3]. Indeed there, the approximated kernel $\widetilde{\mathbf{K}}$ and the ROT distance $W_{\varepsilon,\widetilde{c}}$ associated are not well defined on a neighbourhood of the locations of the inputs measures and therefore are not differentiable.

## 4  Experiments

**Efficiency vs. Approximation trade-off using positive features.** In Figures 1,3 we plot the deviation from ground truth, defined as $D := 100 \times \frac{\text{ROT} - \widetilde{\text{ROT}}}{|\text{ROT}|} + 100$, and show the time-accuracy

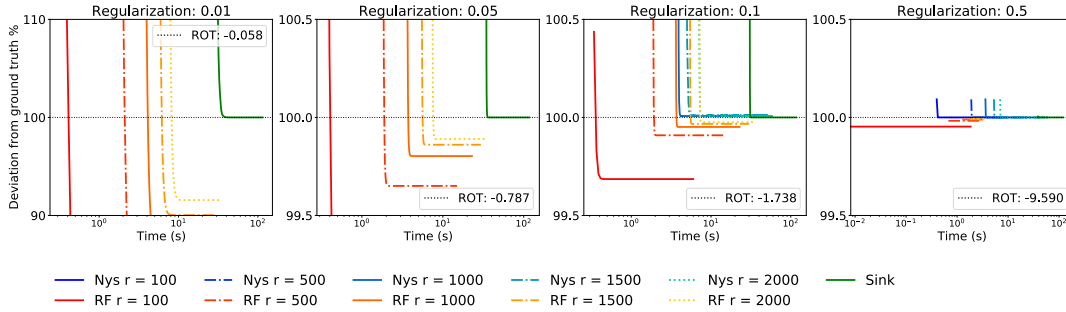

Figure 1: In this experiment, we draw 40000 samples from two normal distributions and we plot the deviation from ground truth for different regularizations. These two normal distributions are in $\mathbb{R}^2$. One of them has mean $(1,1)^T$ and identity covariance matrix $I_2$. The other has 0 mean and covariance $0.1 \times I_2$. We compare the results obtained for our proposed method (**RF**) with the one proposed in [3] (**Nys**) and with the Sinkhorn algorithm (**Sin**) proposed in [16]. The cost function considered here is the square Euclidean metric and the feature map used is that presented in Lemma 1. The number of random features (or rank) chosen varies from 100 to 2000. We repeat for each problem 50 times the experiment. Note that curves in the plot start at different points corresponding to the time required for initialization. *Right*: when the regularization is sufficiently large both **Nys** and **RF** methods obtain very high accuracy with order of magnitude faster than **Sin**. *Middle right, middle left*: **Nys** fails to converge while **RF** works for any given random features and provides very high accuracy of the ROT cost with order of magnitude faster than **Sin**. *Left*: when the regularization is too small all the methods failed as the Nystrom method cannot be computed, the accuracy of the **RF** method is of order of $10\%$ and Sinkhorn algorithm may be too costly.

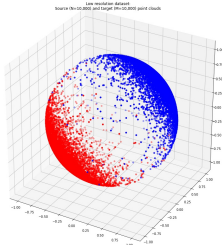

Figure 2: Here we show the two distributions considered in the experiment presented in Figure 3 to compare the time-accuracy tradeoff between the different methods. All the points are drawn on the unit sphere in $\mathbb{R}^3$, and uniform distributions are considered respectively on the red dots and on the blue dots. There are 10000 samples for each distribution.

tradeoff for our proposed method **RF**, Nystrom **Nys** [3] and Sinkhorn **Sin** [16], for a range of regularization parameters $\varepsilon$ (each corresponding to a different ground truth $W_{\varepsilon,c}$) and approximation with $r$ random features in two settings. In particular, we show that our method obtains very high accuracy with order of magnitude faster than **Sin** in a larger regime of regularizations than **Nys**. In Figure 5 in Appendix C, we also show the time-accuracy tradeoff in the high dimensional setting.

**Using positive features to learn adversarial kernels in GANs.** Let $P_X$ a given distribution on $\mathcal{X} \subset \mathbb{R}^D$, $(\mathcal{Z}, \mathcal{A}, \zeta)$ an arbitrary probability space and let $g_\rho : \mathcal{Z} \to \mathcal{X}$ a parametric function where the parameter $\rho$ lives in a topological space $\mathcal{O}$. The function $g_\rho$ allows to generate a distribution on $\mathcal{X}$ by considering the push forward operation through $g_\rho$. Indeed $g_{\rho\sharp}\zeta$ is a distribution on $\mathcal{X}$ and if the function space $\mathcal{F} = \{g_\rho : \rho \in \mathcal{O}\}$ is large enough, we may be able to recover $P_X$ for a well chosen $\rho$. The goal is to learn $\rho^*$ such that $g_{\rho^*_\sharp}\zeta$ is the closest possible to $P_X$ according to a specific metric on the space of distributions. Here we consider the Sinkhorn distance as introduced in Eq.(2). One difficulty when using such metric is to define a well behaved cost to measure the distance between distributions in the ground space. We decide to learn an adversarial cost by embedding the native space $\mathcal{X}$ into a low-dimensional subspace of $\mathbb{R}^d$ thanks to a parametric function $f_\gamma$. Therefore by defining $h_\gamma(x,y) := (f_\gamma(x), f_\gamma(y))$ and given a fixed cost function $c$ on $\mathbb{R}^d$, we can define a parametric cost function on $\mathcal{X}$ as $c \circ h_\gamma(x,y) := c(f_\gamma(x), f_\gamma(y))$. To train a Generative Adversarial Network (GAN), one may therefore optimizes the following objective:

$$\min_\rho \max_\gamma \overline{W}_{\varepsilon, c\circ h_\gamma}(g_{\rho\#}\zeta, P_X)$$

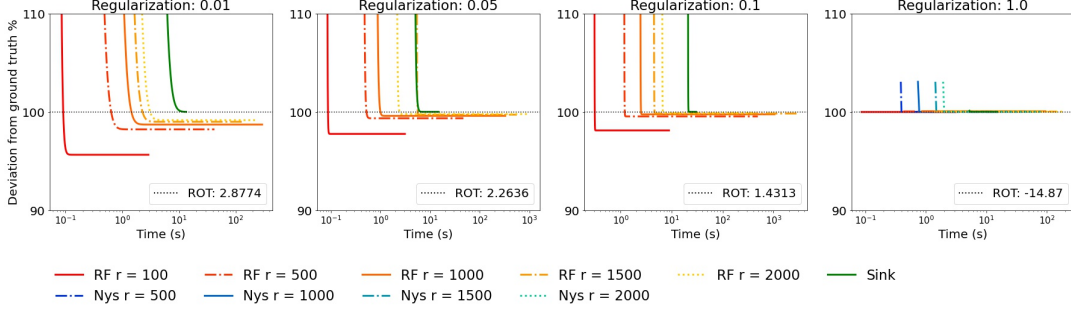

Figure 3: In this experiment, we draw 20000 samples from two distributions on the sphere (see Figure 2) and we plot the deviation from ground truth for different regularizations. We compare the results obtained for our proposed method (**RF**) with the one proposed in [3] (**Nys**) and with the Sinkhorn algorithm (**Sin**) proposed in [16]. The cost function considered here is the square Euclidean metric and the feature map used is that presented in Lemma 1. The number of random features (or rank) chosen varies from 100 to 2000. We repeat for each problem 10 times the experiment. Note that curves in the plot start at different points corresponding to the time required for initialization. *Right*: when the regularization is sufficiently large both **Nys** and **RF** methods obtain very high accuracy with order of magnitude faster than **Sin**. *Middle right, middle left, left*: **Nys** fails to converge while **RF** works for any given random features and provides very high accuracy of the ROT cost with order of magnitude faster than **Sin**.

Indeed, taking the $\max$ of the Sinkhorn distance according to $\gamma$ allows to learn a discriminative cost $c \circ h_\gamma$ [25, 49]. However in practice, we do not have access to the distribution of the data $P_X$, but only to its empirical version $\widehat{P}_X$, where $\widehat{P}_X := \frac{1}{n} \sum_{i=1}^n \delta_{x_i}$ and $\mathbf{X} := \{x_1, ..., x_n\}$ are the $n$ i.i.d samples drawn from $P_X$. By sampling independently $n$ samples $\mathbf{Z} := \{z_1, ..., z_n\}$ from $\zeta$ and denoting $\widehat{\zeta} := \frac{1}{q} \sum_{i=1}^q \delta_{z_i}$ we obtain the following approximation:

$$\min_\rho \max_\gamma \overline{W}_{\varepsilon, c \circ h_\gamma}(g_{\rho\#}\widehat{\zeta}, \widehat{P}_X)$$

However as soon as $n$ gets too large, the above objective, using the classic Sinkhorn Alg. 1 is very costly to compute as the cost of each iteration of Sinkhorn is quadratic in the number of samples. Therefore one may instead split the data and consider $B \geq 1$ mini-batches $\mathbf{Z} = (\mathbf{Z}^b)_{b=1}^B$ and $\mathbf{X} = (\mathbf{X}^b)_{b=1}^B$ of size $s = \frac{n}{B}$, and obtain instead the following optimisation problem:

$$\min_\rho \max_\gamma \frac{1}{B} \sum_{b=1}^B \overline{W}_{\varepsilon, c \circ h_\gamma}(g_{\rho\#}\widehat{\zeta}^b, \widehat{P}_X^b)$$

where $\widehat{\zeta}^b := \frac{1}{s} \sum_{i=1}^s \delta_{z_i^b}$ and $\widehat{P}_X^b := \frac{1}{s} \sum_{i=1}^s \delta_{x_i^b}$. However the smaller the batches are, the less precise the approximation of the objective is. To overcome this issue we propose to apply our method and replace the cost function $c$ by an approximation defined as $c_\theta(x, y) = -\epsilon \log \varphi_\theta(x)^T \varphi_\theta(y)$ and consider instead the following optimisation problem:

$$\min_\rho \max_\gamma \frac{1}{B} \sum_{b=1}^B \overline{W}_{\varepsilon, c_\theta \circ h_\gamma}(g_{\rho\#}\widehat{\zeta}^b, \widehat{P}_X^b).$$

Indeed in that case, the Gibbs kernel associated to the cost function $c_\theta \circ h_\gamma$ is still factorizafable as we have $c_\theta \circ h_\gamma(x, y) = -\epsilon \log \varphi_\theta(f_\gamma(x))^T \varphi_\theta(f_\gamma(y))$. Such procedure allows us to compute the objective in linear time and therefore to largely increase the size of the batches. Note that we keep the batch formulation as we still need it because of memory limitation on GPUs. Moreover, we may either consider a random approximation by drawing $\theta$ randomly for a well chosen distribution or we could learn the random features $\theta$. In the following we decide to learn the features $\theta$ in order to obtain a cost function $c_\theta \circ h_\gamma$ even more discriminative. Finally our objective is:

$$\min_\rho \max_{\gamma, \theta} \frac{1}{B} \sum_{b=1}^B \overline{W}_{\varepsilon, c_\theta \circ h_\gamma}(g_{\rho\#}\widehat{\zeta}^b, \widehat{P}_X^b) \tag{18}$$

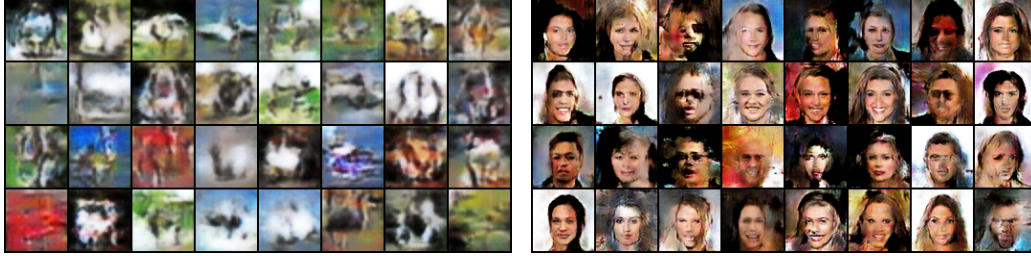

Figure 4: Images generated by two learned generative models trained by optimizing the objective (18) where we set the number of batches $s = 7000$, the regularization $\varepsilon = 1$, and the number of features $r = 600$. *Left, right:* samples obtained from the proposed generative model trained on respectively CIFAR-10 [35] and celebA [38].

Therefore here we aim to learn an embedding from the input space into the feature space thanks to two operations. The first one consists in taking a sample and embedding it into a latent space thanks to the mapping $f_\gamma$ and the second one is an embedding of this latent space into the feature space thanks to the feature map $\varphi_\theta$. From now on we assume that $g_\rho$ and $f_\gamma$ are neural networks. More precisely we take the exact same functions used in [46, 36] to define $g_\rho$ and $f_\gamma$. Moreover, $\varphi_\theta$ is the feature map associated to the Gaussian kernel defined in Lemma 1 where $\theta$ is initialised with a normal distribution. The number of random features considered has been fixed to be $r = 600$ in the following. The training procedure is the same as [27, 36] and consists in alternating $n_c$ optimisation steps to train the cost function $c_\theta \circ h_\gamma$ and an optimisation step to train the generator $g_\rho$. The code is available at github.com/meyerscetbon/LinearSinkhorn.

| $k_\theta(f_\gamma(x), f_\gamma(z))$ | Image $x$ | Noise $z$ |
|---|---|---|
| Image $x$ | $1802 \times 1e12$ | $2961 \times 1e5$ |
| Noise $z$ | $2961 \times 1e5$ | $48.65$ |

Table 1: Comparison of the learned kernel $k_\theta$, trained on CIFAR-10 by optimizing the objective (18), between images taken from CIFAR-10 and random noises sampled in the native of space of images. The values shown are averages obtained between 5 noise and/or image samples. As we can see the cost learned has well captured the structure of the image space.

**Optimisation.** Thanks to proposition 3.2, the objective is differentiable with respect to $\theta, \gamma$ and $\rho$. We obtain the gradient by computing an approximation of the gradient thanks to the approximate dual variables obtained by the Sinkhorn algorithm. We refers to section 3.3 for the expression of the gradient. This strategy leads to two benefits. First it is memory efficient as the computation of the gradient at this stage does not require to keep track of the computations involved in the Sinkhorn algorithm. Second it allows, for a given regularization, to compute with very high accuracy the Sinkhorn distance. Therefore, our method may be applied also for small regularization.

**Results.** We train our GAN models on a Tesla K80 GPU for 84 hours on two different datasets, namely CIFAR-10 dataset [35] and CelebA dataset [38] and learn both the proposed generative model and the adversarial cost function $c_\theta$ derived from the adversarial kernel $k_\theta$. Figure 4 illustrates the generated samples and Table 1 displays the geometry captured by the learned kernel.

**Discussion.** Our proposed method has mainly two advantages compared to the other Wasserstein GANs (W-GANs) proposed in the literature. First, the computation of the Sinkhorn divergence is linear with respect to the number of samples which allow to largely increase the batch size when training a W-GAN and obtain a better approximation of the true Sinkhorn divergence. Second, our approach is fully differentiable and therefore we can directly compute the gradient of the Sinhkorn divergence with respect the parameters of the network. In [49] the authors do not differentiate through the Wasserstein cost to train their network. In [25] the authors do differentiate through the iterations of the Sinkhorn algorithm but this strategy require to keep track of the computation involved in the Sinkhorn algorithm and can be applied only for large regularizations as the number of iterations cannot be too large.

## Acknowledgements

This work was funded by a "Chaire d'excellence de l'IDEX Paris Saclay".

## Broader Impact

Optimal Transport (OT) has gained interest last years in machine learning with many applications in neuroimaging, generative models, supervised learning, word embeddings, reconstruction cell trajectories or adversarial examples. This work brings new applications to OT in the high dimensional setting as it provides a linear time method to compute an approximation of the OT cost and gives a constructive method to learn an adapted kernel or equivalently an adapted cost function depending on the problem considered.

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
