[Supplementary Material]

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

## Supplementary materials

**Outline.** In Sec. A we provide the proofs related to the approximation proprieties of our proposed method. In Sec. B we show the differentiability of the constructive approach. Finally in Sec. C we add more experiments and illustrations of our proposed method.

## A  Approximation via Random Fourier Features

### A.1  Proof of Theorem 3.1

In the following we denote $\mathbf{K} = (k(x_i, y_j))_{i,j=1}^{n}$ $\mathbf{K}_\theta = (k_\theta(x_i, y_j))_{i,j=1}^{n}$ the two gram matrices associated with $k$ and $k_\theta$ respectively. By duality and from these two matrices we can define the two objectives to maximize to obtain $W_{\varepsilon,c}$ and $W_{\varepsilon,c_\theta}$:

$$W_{\varepsilon,c} = \max_{\alpha,\beta} f(\alpha,\beta) := \langle \alpha, a \rangle + \langle \beta, b \rangle - \varepsilon \langle e^{\alpha/\varepsilon}, \mathbf{K} e^{\beta/\varepsilon} \rangle$$

$$W_{\varepsilon,c_\theta} = \max_{\alpha,\beta} f_\theta(\alpha,\beta) := \langle \alpha, a \rangle + \langle \beta, b \rangle - \varepsilon \langle e^{\alpha/\varepsilon}, \mathbf{K}_\theta e^{\beta/\varepsilon} \rangle$$

Moreover as $k$ and $\varphi$ are assumed to be positive, there exists unique (up to a scalar translation) $(\alpha^*, \beta^*)$ and $(\alpha_\theta^*, \beta_\theta^*)$ respectively solutions of $\max_{\alpha,\beta} f(\alpha,\beta)$ and $\max_{\alpha,\beta} f_\theta(\alpha,\beta)$.

**Proof.** *Let us first show the following proposition:*

**Proposition 1.** *Let $\delta > 0$ and $r \geq 1$. Assume that for all $(x,y) \in \mathbf{X} \times \mathbf{Y}$,*

$$\left| \frac{k(x,y) - k_\theta(x,y)}{k(x,y)} \right| \leq \frac{\delta \varepsilon^{-1}}{2 + \delta \varepsilon^{-1}} \tag{19}$$

*then Sinkhorn Alg. 1 with inputs $a, b, K_\theta$ outputs $(\alpha_\theta, \beta_\theta)$ in*

$$\mathcal{O}\left( \frac{nr}{\delta \varepsilon^{-1}} \left[ \log\left(\frac{1}{\iota}\right) + \log\left(2 + \delta \varepsilon^{-1}\right) + \varepsilon^{-1} R^2 \right]^2 \right)$$

*where*

$$\iota = \min_{i,j}(a_i, b_j) \quad \text{and} \quad R = \max_{(x,y) \in \mathbf{X} \times \mathbf{Y}} c(x,y). \tag{20}$$

*such that:*

$$|W_{\varepsilon,c} - f_\theta(\alpha_\theta, \beta_\theta)| \leq \delta$$

**Proof.** *We remark that:*

$$|f(\alpha^*, \beta^*) - f_\theta(\alpha_\theta, \beta_\theta)| \leq |f(\alpha^*, \beta^*) - f(\alpha_\theta^*, \beta_\theta^*)|$$
$$+ |f(\alpha_\theta^*, \beta_\theta^*) - f_\theta(\alpha_\theta^*, \beta_\theta^*)|$$
$$+ |f_\theta(\alpha_\theta^*, \beta_\theta^*) - f_\theta(\alpha_\theta, \beta_\theta)|$$

*Moreover we have that:*

$$|f(\alpha^*, \beta^*) - f(\alpha_\theta^*, \beta_\theta^*)| = f(\alpha^*, \beta^*) - f(\alpha_\theta^*, \beta_\theta^*)$$
$$= f(\alpha^*, \beta^*) - f_\theta(\alpha_\theta^*, \beta_\theta^*) + f_\theta(\alpha_\theta^*, \beta_\theta^*) - f(\alpha_\theta^*, \beta_\theta^*)$$
$$\leq |f(\alpha^*, \beta^*) - f_\theta(\alpha^*, \beta^*)| + |f_\theta(\alpha_\theta^*, \beta_\theta^*) - f(\alpha_\theta^*, \beta_\theta^*)|$$

*Therefore we obtain that:*

$$|f(\alpha^*, \beta^*) - f_\theta(\alpha_\theta, \beta_\theta)| \leq 2|f(\alpha_\theta^*, \beta_\theta^*) - f_\theta(\alpha_\theta^*, \beta_\theta^*)| + |f(\alpha^*, \beta^*) - f_\theta(\alpha^*, \beta^*)|$$
$$+ |f_\theta(\alpha_\theta^*, \beta_\theta^*) - f_\theta(\alpha_\theta, \beta_\theta)|$$

*Let us now introduce the following lemma:*

**Lemma 2.** *Let $1 > \tau > 0$ and let us assume that for all $(x,y) \in \mathbf{X} \times \mathbf{Y}$,*

$$\left| \frac{k(x,y) - k_\theta(x,y)}{k(x,y)} \right| \leq \tau$$

*then for any $\alpha, \beta \in \mathbb{R}^n$ it holds*

$$|f(\alpha, \beta) - f_\theta(\alpha, \beta)| \leq \varepsilon \tau [\langle e^{\varepsilon^{-1}\alpha}, \mathbf{K}e^{\varepsilon^{-1}\beta} \rangle] \tag{21}$$

*and*

$$|f(\alpha, \beta) - f_\theta(\alpha, \beta)| \leq \varepsilon \frac{\tau}{1-\tau} [\langle e^{\varepsilon^{-1}\alpha}, \mathbf{K}_\theta e^{\varepsilon^{-1}\beta} \rangle] \tag{22}$$

**Proof.** *Let $\alpha, \beta \in \mathbb{R}^n$. We remarks that:*

$$f(\alpha, \beta) - f_\theta(\alpha, \beta) = \varepsilon [\langle e^{\varepsilon^{-1}\alpha}, (\mathbf{K}_\theta - \mathbf{K})e^{\varepsilon^{-1}\beta} \rangle] \tag{23}$$

*Therefore we obtain that:*

$$|f(\alpha, \beta) - f_\theta(\alpha, \beta)| \leq \varepsilon \sum_{i,j=1}^{n} e^{\varepsilon^{-1}\alpha_i} e^{\varepsilon^{-1}\beta_j} |[\mathbf{K}_\theta]_{i,j} - \mathbf{K}_{i,j}| \tag{24}$$

*And the first inequality follows from the fact that $|[\mathbf{K}_\theta]_{i,j} - \mathbf{K}_{i,j}| \leq \tau |\mathbf{K}_{i,j}|$ for all $i, j \in \{1, ..., n\}$ and that $k$ is positive. Moreover from the same inequality we obtain that:*

$$|[\mathbf{K}_\theta]_{i,j} - \mathbf{K}_{i,j}| \leq \frac{\tau}{1-\tau} [\mathbf{K}_\theta]_{i,j}$$

*Therefore the second inequality follows.*

*Therefore thanks to lemma 2, we obtain that:*

$$|f(\alpha_\theta^*, \beta_\theta^*) - f_\theta(\alpha_\theta^*, \beta_\theta^*)| \leq \varepsilon \frac{\tau}{1-\tau} [\langle e^{\varepsilon^{-1}\alpha_\theta^*}, \mathbf{K}_\theta e^{\varepsilon^{-1}\beta_\theta^*} \rangle] \tag{25}$$

*But as $(\alpha_\theta^*, \beta_\theta^*)$ is the optimum of $f_\theta$, the first order conditions give us that $\langle e^{\varepsilon^{-1}\alpha_\theta^*}, \mathbf{K}_\theta e^{\varepsilon^{-1}\beta_\theta^*} \rangle = 1$ and finally we have:*

$$|f(\alpha_\theta^*, \beta_\theta^*) - f_\theta(\alpha_\theta^*, \beta_\theta^*)| \leq \varepsilon \frac{\tau}{1-\tau} \tag{26}$$

*Thanks to lemma 2, we also deduce that:*

$$|f(\alpha^*, \beta^*) - f_\theta(\alpha^*, \beta^*)| \leq \varepsilon \tau \tag{27}$$

*Let us now introduce the following theorem:*

**Theorem A.1.** *([19]) Given $\mathbf{K}_\theta \in \mathbb{R}^{n \times n}$ with positive entries and $a, b \in \Delta_n$ the Sinkhorn Alg. 1 computes $(\alpha_\theta, \beta_\theta)$ such that*

$$|f_\theta(\alpha_\theta^*, \beta_\theta^*) - f_\theta(\alpha_\theta, \beta_\theta)| \leq \frac{\delta}{2}$$

*in $\mathcal{O}\left(\delta^{-1}\varepsilon \log\left(\frac{1}{\iota \min_{i,j}[K_\theta]_{i,j}}\right)^2\right)$ iterations where $\iota = \min_{i,j}(a_i, b_j)$ and each of which requires $\mathcal{O}(1)$ matrix-vector products with $K_\theta$ and $\mathcal{O}(n)$ additional processing time.*

*Moreover from Eq. (19) we have that*

$$[\mathbf{K}_\theta]_{i,j} \geq (1-\tau)\mathbf{K}_{i,j} \tag{28}$$

*where $\tau = \frac{\delta \varepsilon^{-1}}{2 + \delta \varepsilon^{-1}}$, therefore $\log\left(\frac{1}{\min_{i,j}[\mathbf{K}_\theta]_{i,j}}\right) \leq \log\left(\frac{1}{(1-\tau)\min_{i,j}\mathbf{K}_{i,j}}\right) \leq \log\left(\frac{1}{1-\tau}\right) + \varepsilon^{-1}R^2$ where $R = \max_{(x,y) \in \mathbf{X} \times \mathbf{Y}} c(x, y)$ and we obtain that*

$$|f(\alpha^*, \beta^*) - f_\theta(\alpha_\theta, \beta_\theta)| \leq 2\varepsilon \frac{\tau}{1-\tau} + \varepsilon \tau + \frac{\delta}{2} \tag{29}$$

*By replacing $\tau$ by its value, we obtain the desired result.*

We are now ready to prove the theorem. Let $r \geq 1$. From theorem A.1, we obtain directly that:

$$|f(\alpha^*, \beta^*) - f_\theta(\alpha_\theta, \beta_\theta)| \leq \frac{\delta}{2} \tag{30}$$

in $\mathcal{O}\left(\frac{nr}{\delta}\left[\log\left(\frac{1}{\iota}\right) + Q_\theta\right]^2\right)$ algebric operations. Moreover let $\tau > 0$ and

$$r \in \Omega\left(\frac{\psi^2}{\delta^2}\left[\min\left(d\varepsilon^{-1}R^2 + d\log\left(\frac{\psi VD}{\tau\delta}\right), \log\left(\frac{n}{\tau}\right)\right)\right]\right)$$

and $u_1, ..., u_r$ drawn independently from $\rho$. Then from Proposition 3.1 we obtain that with a probability of at least $1 - \delta$ it holds for all $(x, y) \in \mathbf{X} \times \mathbf{Y}$,

$$\left|\frac{k(x,y) - k_\theta(x,y)}{k(x,y)}\right| \leq \frac{\delta\varepsilon^{-1}}{2 + \delta\varepsilon^{-1}} \tag{31}$$

and the result follows from Proposition 1.

## A.2  Accelerated Version

[31] show that one can accelarated the Sinkhorn algorithm (see Alg. 2) and obtain a $\delta$-approximation of the ROT distance. For that purpose, [31] introduce a reformulation of the dual problem (8) and obtain

$$W_{\varepsilon,c_\theta} = \sup_{\eta_1,\eta_2} F_\theta(\eta_1, \eta_2) := \varepsilon\left[\langle\eta_1, a\rangle + \langle\eta_2, b\rangle - \log\left(\langle\mathbf{K}_\theta e^{\eta_2}\rangle\right)\right] \tag{32}$$

which can be shown to be an $L$-smooth function ([41]) where $L \leq 2\varepsilon^{-1}$. Let us now present our result using the accelarated Sinkhorn algorithm.

**Theorem A.2.** *Let $\delta > 0$ and $r \geq 1$. Then the Accelerated Sinkhorn Alg. 2 with inputs $\mathbf{K}_\theta$, $a$ and $b$ outputs $(\alpha_\theta, \beta_\theta)$ such that*

$$|W_{\varepsilon,c_\theta} - F_\theta(\alpha_\theta, \beta_\theta)| \leq \frac{\delta}{2}$$

*in $\mathcal{O}\left(\frac{nr}{\sqrt{\delta}}[\sqrt{\varepsilon^{-1}}A_\theta]\right)$ algebraic operations where $A_\theta = \inf_{(\alpha,\beta)\in\Theta_\theta}\|(\alpha,\beta)\|_2$ and $\Theta_\theta$ is the set of optimal dual solutions of (8). Moreover let $\tau > 0$,*

$$r \in \Omega\left(\frac{\psi^2}{\delta^2}\left[\min\left(d\varepsilon^{-1}\|C\|_\infty^2 + d\log\left(\frac{\psi VD}{\delta\delta}\right), \log\left(\frac{n}{\delta}\right)\right)\right]\right) \tag{33}$$

*and $u_1, ..., u_r$ drawn independently from $\rho$, then with a probability $1 - \tau$ it holds*

$$|W_{\varepsilon,c} - F_\theta(\alpha_\theta, \beta_\theta)| \leq \delta \tag{34}$$

**Proof.** *Let us first introduce the theorem presented in [31]:*

**Theorem A.3.** *Given $\mathbf{K}_\theta \in \mathbb{R}^{n \times n}$ with positive entries and $a, b \in \Delta_n$ the Accelerated Sinkhorn Alg. (2) computes $(\alpha_\theta, \beta_\theta)$ such that*

$$|W_{\varepsilon,c_\theta} - F_\theta(\alpha_\theta, \beta_\theta)| \leq \delta$$

*in $\mathcal{O}\left(\sqrt{\frac{\eta}{\delta}}A_\theta\right)$ iterations where $A_\theta = \inf_{(\alpha_\theta^*, \beta_\theta^*)\in\Theta^*}\|(\alpha_\theta^*, \beta_\theta^*)\|_2$ and $\Theta^*$ is the set of optimal dual solutions. Moreover each of which requires $\mathcal{O}(1)$ matrix-vector products with $\mathbf{K}_\theta$ and $\mathcal{O}(n)$.*

*From the above result and applying an analogue proof of Theorem A.1, we obtain the desired result.*

## A.3  Proof of Proposition 3.1

**Proof.** *The proof is given for $p = 1$ but it hold also for any $p \geq 1$ after making some simple modifications. To obtain the first inequality we remarks that*

$$\mathbb{P}\left(\sup_{(x,y)\in\mathcal{X}\times\mathcal{X}}\left|\frac{k_\theta(x,y)}{k(x,y)} - 1\right| \geq \delta\right) \leq \sum_{(x,y)\in\mathbf{X}\times\mathbf{Y}}\mathbb{P}\left(\left|\frac{k_\theta(x,y)}{k(x,y)} - 1\right| \geq \delta\right) \tag{35}$$

*Moreover as $\mathbf{E}_\rho\left(\frac{\varphi(x,u)\varphi(y,u)}{k(x,y)}\right) = 1$, the result follows by applying Hoeffding's inequality.*

**Algorithm 2** Accelerated Sinkhorn Algorithm.

---

**Input:** Initial estimate of the Lipschitz constant $L_0$, $a$, $b$, and $\mathbf{K}$
**Init:** $A_0 = \alpha_0 = 0$, $\eta^0 = \zeta^0 = \lambda^0 = 0$.
**for** $k \geq 0$ **do**
$\quad L_{k+1} = L_k/2$
$\quad$**while** *True* **do**
$\quad\quad$ Set $L_{k+1} = L_k/2$
$\quad\quad$ Set $a_{k+1} = \frac{1}{2L_{k+1}} + \sqrt{\frac{1}{4L_{k+1}^2} + a_k^2 \frac{L_k}{L_{k+1}}}$
$\quad\quad$ Set $\tau_k = \frac{1}{a_{k+1}L_{k+1}}$
$\quad\quad$ Set $\lambda^k = \tau_k \zeta^k + (1 - \tau_k)\zeta^k$
$\quad\quad$ Choose $i_k = \underset{i \in \{1,2\}}{\operatorname{argmax}} \|\nabla_i \phi(\lambda^k)\|_2$
$\quad\quad$**if** $i_k = 1$ **then**
$\quad\quad\quad \eta_1^{k+1} = \lambda_1^k + \log(a) - \log(e^{\lambda_1^k} \circ \mathbf{K} e^{\lambda_2^k})$
$\quad\quad\quad \eta_2^{k+1} = \lambda_2^{k+1}$
$\quad\quad\quad$**else**
$\quad\quad\quad\quad \eta_1^{k+1} = \lambda_1^{k+1}$
$\quad\quad\quad\quad \eta_2^{k+1} = \lambda_2^k + \log(b) - \log(e^{\lambda_2^k} \circ \mathbf{K}^T e^{\lambda_1^k})$
$\quad\quad\quad$**end**
$\quad\quad$**end**
$\quad\quad$ Set $\zeta^{k+1} = \zeta^k - a_{k+1}\nabla F_\theta(\lambda^k)$
$\quad\quad$**if** $\phi(\eta^k + 1) \leq \phi(\lambda^k) - \frac{\|\nabla F_\theta(\lambda^k)\|^2}{2L_{k+1}}$ **then**
$\quad\quad\quad$ Set $z = \operatorname{Diag}(e^{\lambda_1^k}) \circ \mathbf{K} \circ \operatorname{Diag}(e^{\lambda_2^k})$
$\quad\quad\quad$ Set $c = \langle e^{\lambda_1^k}, \mathbf{K} e^{\lambda_2^k} \rangle$
$\quad\quad\quad$ Set $\hat{x}^{k+1} = \frac{a_{k+1}c^{-1}z + L_k a_k^2 \hat{x^k}}{L_{k+1}a_{k+1}^2}$
$\quad\quad\quad$**Break**
$\quad\quad$**end**
$\quad\quad$ Set $L_{k+1} = 2L_{k+1}$
$\quad$**end**
**end**
**Result**: Transport Plan $\hat{x}^{k+1}$ and dual points $\eta^{k+1} = (\eta_1^{k+1}, \eta_2^{k+1})^T$

---

*To show the second inequality, we follow the same strategy adopted in [47]. Let us denote $f(x,y) = \frac{k_\theta(x,y)}{k(x,y)} - 1$ and $\mathcal{M} := \mathcal{X} \times \mathcal{X}$. First we remarks that $|f(x,y)| \leq K + 1$ and $\mathbf{E}_\rho(f) = 0$. As $\mathcal{M}$ is a compact, we can find an $\mu$-net that covers $\mathcal{M}$ with $\mathcal{N}(\mathcal{M}, \mu) = \left(\frac{4R}{\mu}\right)^{2d}$ where $R = \sup_{(x,y)} \|(x,y)\|_2$ balls of radius $\delta$. Let us denote $z_1, ..., z_{\mathcal{N}(\mathcal{M},\mu)} \in \mathcal{M}$ the centers of these balls, and let $L_f$ denote the Lipschitz constant of $f$. As $f$ is differentiable We have therefore $L_f = \sup_{z \in \mathcal{M}} \|\nabla f(z)\|_2$. Moreover we have:*

$$\nabla f(z) = \frac{\nabla k_\theta(z)}{k(z)} - \frac{k_\theta(z)}{k(z)}\nabla k(z) \tag{36}$$

$$= \frac{1}{k(z)}\left[(\nabla k_\theta(z) - \nabla k(z)) + \nabla k(z)\left(1 - \frac{k_\theta(z)}{k(z)}\right)\right] \tag{37}$$

*Therefore we have*

$$\mathbf{E}(\|\nabla f(z)\|^2) \leq \frac{2}{k(z)^2}\left[\mathbf{E}(\|\nabla k_\theta(z) - \nabla k(z)\|^2) + \|\nabla k(z)\|^2 \mathbf{E}\left(1 - \frac{k_\theta(z)}{k(z)}\right)^2\right] \tag{38}$$

*But for any $z \in \mathcal{M}$ we have from Eq. (15) :*

$$\mathbf{E}\left(1 - \frac{k_\theta(z)}{k(z)}\right)^2 = \int_{t \geq 0} \mathbb{P}\left(\left(1 - \frac{k_\theta(z)}{k(z)}\right)^2 \geq t\right) \tag{39}$$

$$\leq \frac{K^2}{r} \tag{40}$$

*Moreover, we have:*

$$\nabla k_\theta(z) = \frac{1}{r}\sum_{i=1}^{r} \nabla_x \varphi(x, u_i)\varphi(y, u_i) + \varphi(x, u_i)\nabla_y \varphi(y, u_i) \tag{41}$$

*Therefore we have:*

$$\|\nabla k_\theta(z)\|^2 = \frac{1}{r^2}\sum_{i,j=1}^{r} \langle \nabla_x \varphi(x, u_i), \nabla_x \varphi(x, u_j)\rangle \varphi(y, u_i)\varphi(y, u_j)$$

$$+ \frac{1}{r^2}\sum_{i,j=1}^{r} \nabla_y \varphi(y, u_i), \nabla_y \varphi(y, u_j)\rangle \varphi(x, u_i)\varphi(x, u_j)$$

$$+ \frac{2}{r^2}\sum_{i,j=1}^{r} \nabla_x \varphi(x, u_i), \nabla_y \varphi(x, u_j)\rangle \varphi(y, u_i)\varphi(x, u_j)$$

*Moreover as:*

$$|\varphi(y, u_i)\varphi(x, u_j)| \leq \frac{\varphi(y, u_i)^2 + \varphi(x, u_j)^2}{2} \tag{42}$$

$$\leq K \sup_{x \in \mathcal{X}} k(x, x) \tag{43}$$

*And:*

$$|\langle \nabla_x \varphi(x, u_i), \nabla_y \varphi(y, u_j)\rangle| \leq \|\nabla_x \varphi(x, u_i)\|\|\nabla_y \varphi(y, u_j)\| \tag{44}$$

$$\leq \frac{\|\nabla_x \varphi(x, u_i)\|^2 + \|\nabla_y \varphi(y, u_j)\|^2}{2} \tag{45}$$

*And by denoting:*

$$V := \sup_{x \in \mathcal{X}} \mathbf{E}_\rho \left(\|\nabla_x \varphi(x, u)\|^2\right) \tag{46}$$

*Therefore we have:*

$$\mathbf{E}\left(|\langle \nabla_x \varphi(x, u_i), \nabla_y \varphi(y, u_j)\rangle|\right) \leq V \tag{47}$$

*We can now derive the following upper bound:*

$$\mathbf{E}(\|\nabla k_\theta(z) - \nabla k(z)\|^2) = \mathbf{E}(\|\nabla k_\theta(z)\|^2) - \|\nabla k(z)\|^2 \leq 4VK \sup_{x \in \mathcal{X}} k(x, x) \tag{48}$$

*Moreover by convexity of the $\ell_2$ square norm, we also obtain that:*

$$\|\nabla k(z)\|^2 \leq VK \sup_{x \in \mathcal{X}} k(x, x) \tag{49}$$

*Therefore we have*

$$\mathbf{E}(\|\nabla f(z)\|^2) \leq 2\kappa^{-2}VK \sup_{x \in \mathcal{X}} k(x, x)\left[4 + \frac{K^2}{r}\right] \tag{50}$$

*Then by applying Markov inequality we obtain that:*

$$\mathbb{P}\left(L_f \geq \frac{\delta}{2\mu}\right) \leq 2\kappa^{-2}VK \sup_{x \in \mathcal{X}} k(x, x)\left[4 + \frac{K^2}{r}\right]\left(\frac{2\mu}{\delta}\right)^2 \tag{51}$$

*Moreover, the union bound followed by Hoeffding's inequality applied to the anchors in the $\mu$-net gives*

$$\mathbb{P}\left(\cup_{i=1}^{\mathcal{N}(\mathcal{M},\mu)}|f(z_i)| \geq \delta\right) \leq 2\mathcal{N}(\mathcal{M},\mu)\exp\left(-\frac{r\delta^2}{2K^2}\right) \tag{52}$$

*Then by combining Eq. (51) and Eq.(52) we obtain that:*

$$\mathbb{P}\left(\sup_{z\in\mathcal{M}}|f(z)| \geq \delta\right) \leq 2\left(\frac{4R}{\mu}\right)^{2d}\exp\left(-\frac{r\delta^2}{2K^2}\right) + 2\kappa^{-2}VK\sup_{x\in\mathcal{X}}k(x,x)\left[4+\frac{K^2}{r}\right]\left(\frac{2\mu}{\delta}\right)^2$$

*Therefore by denoting*

$$A_1 := 2\left(4R\right)^{2d}\exp\left(-\frac{r\delta^2}{2K^2}\right) \tag{53}$$

$$A_2 := 2\kappa^{-2}VK\sup_{x\in\mathcal{X}}k(x,x)\left[4+\frac{K^2}{r}\right]\left(\frac{2}{\delta}\right)^2 \tag{54}$$

*and by choosing $\mu = \frac{A_1}{A_2}^{\frac{1}{2d+2}}$, we obtain that:*

$$\mathbb{P}\left(\sup_{z\in\mathcal{M}}|f(z)| \geq \delta\right) \leq 2^9\left[\frac{\kappa^{-2}KV\sup_{x\in\mathcal{X}}k(x,x)\left[4+\frac{K^2}{r}\right]R^2}{\delta^2}\right]\exp\left(-\frac{r\delta^2}{2K^2(d+1)}\right)$$

**Ratio Approximation.** Let us assume here that $p = 1$ for simplicity. The uniform bound obtained on the ratio gives naturally a control of the form Eq.(14) with a prescribed number of random features $r$. This result allows to control the error when using the kernel matrix $\mathbf{K}_\theta$ instead of the true kernel matrix $\mathbf{K}$ in the Sinkhorn iterations. In the proposition above, we obtain such a result with a probability of at least $1 - 2n^2\exp\left(-\frac{r\delta^2}{2\psi^2}\right)$ where $r$ is the number of random features and $\psi$ is defined as

$$\psi := \sup_{u\in\mathcal{U}}\sup_{(x,y)\in\mathbf{X}\times\mathbf{Y}}\left|\frac{\varphi(x,u)\varphi(y,u)}{k(x,y)}\right|.$$

In comparison, in [47], the authors obtain a uniform bound on their difference and by denoting

$$\phi = \sup_{u\in\mathcal{U}}\sup_{(x,y)\in\mathbf{X}\times\mathbf{Y}}|\varphi(x,u)\varphi(y,u)|,$$

one obtains that with a probability of at least $1 - 2n^2\exp\left(-\frac{r\tau^2}{2\phi^2}\right)$ for all $(x,y)\in\mathbf{X}\times\mathbf{Y}$

$$k(x,y) - \tau \leq k_\theta(x,y) \leq k(x,y) + \tau \tag{55}$$

To be able to recover Eq.(14) from the above control, we need to take $\tau = \inf_{x,y\in\mathbf{X}\times\mathbf{Y}}k(x,y)\delta$ and by denoting $\phi' = \frac{\phi}{\inf_{x,y\in\mathbf{X}\times\mathbf{Y}}k(x,y)}$ we obtain that with a probability of at least $1 - 2n^2\exp\left(-\frac{r\delta^2}{2\phi'^2}\right)$ for all $(x,y)\in\mathbf{X}\times\mathbf{Y}$

$$(1-\delta)k(x,y) \leq k_\theta(x,y) \leq (1+\delta)k(x,y)$$

Therefore the number of random features needed to guarantee Eq.(14) from a control between the difference of the two kernels with at least a probability $1 - \delta$ has to be larger than $\left(\frac{\phi'}{\psi}\right)^2$ times the number of random features needed from the control of Proposition 3.1 to guarantee Eq.(14) with at least the same probability $1 - \delta$. But we always have that

$$\psi = \sup_{u\in\mathcal{U}}\sup_{(x,y)\in\mathbf{X}\times\mathbf{Y}}\left|\frac{\varphi(x,u)\varphi(y,u)}{k(x,y)}\right| \leq \frac{\sup_{u\in\mathcal{U}}\sup_{(x,y)\in\mathbf{X}\times\mathbf{Y}}|\varphi(x,u)\varphi(y,u)|}{\inf_{x,y\in\mathbf{X}\times\mathbf{Y}}k(x,y)} = \phi'$$

and in some cases the ratio $\left(\frac{\phi'}{\psi}\right)^2$ can be huge. Indeed, as we will see in the following, for the Gaussian kernel,

$$k(x, y) = \exp(-\varepsilon^{-1}\|x - y\|_2^2)$$

there exists $\varphi$ and $\mathcal{U}$ such that for all $x, y$ and $u \in \mathcal{U}$:

$$\varphi(x, u)\varphi(y, u) = k(x, y)h(u, x, y)$$

where for all $(x_0, y_0) \in \mathbf{X} \times \mathbf{Y}$,

$$\sup_{u \in \mathcal{U}} |h(u, x_0, y_0)| = \sup_{u \in \mathcal{U}} \sup_{(x,y) \in \mathbf{X} \times \mathbf{Y}} |h(u, x, y)|.$$

Therefore by denoting $M = \sup_{(x,y) \in \mathbf{X} \times \mathbf{Y}} \|x - y\|_2$ and $m = \inf_{(x,y) \in \mathbf{X} \times \mathbf{Y}} \|x - y\|_2$ , we obtain that

$$\left(\frac{\phi'}{\psi}\right)^2 = \left(\frac{\sup_{x,y \in \mathbf{X} \times \mathbf{Y}} k(x, y)}{\inf_{x,y \in \mathbf{X} \times \mathbf{Y}} k(x, y)}\right)^2 = \exp\left(2\varepsilon^{-1}[M^2 - m^2]\right)$$

### A.4 Proof of Lemma 1

**Proof.** *Let $\varepsilon > 0$ and $x, y \in \mathbb{R}^d$. We have that:*

$$\exp\left(-2\varepsilon^{-1}\|x - u\|_2^2\right)\exp\left(-2\varepsilon^{-1}\|y - u\|_2^2\right) = \exp\left(-\varepsilon^{-1}\|x - y\|_2^2\right)\exp\left(-4\varepsilon^{-1}\left\|u - \left(\frac{x + y}{2}\right)\right\|_2^2\right)$$

$$(56)$$

*And as the LHS is integrable we have:*

$$\int_{u \in \mathbb{R}^d} \exp\left(-2\varepsilon^{-1}\|x - u\|_2^2\right)\exp\left(-2\varepsilon^{-1}\|y - u\|_2^2\right)du = \int_{u \in \mathbb{R}^d} e^{-\varepsilon^{-1}\|x - y\|_2^2}\exp\left(-4\varepsilon^{-1}\left\|u - \left(\frac{x + y}{2}\right)\right\|_2^2\right)du$$

*Therefore we obtain that:*

$$e^{-\varepsilon^{-1}\|x - y\|_2^2} = \left(\frac{4}{\pi\varepsilon}\right)^{d/2}\int_{u \in \mathbb{R}^d} \exp\left(-2\varepsilon^{-1}\|x - u\|_2^2\right)\exp\left(-2\varepsilon^{-1}\|y - u\|_2^2\right)du \qquad (57)$$

*Now we want to transform the above expression as the one stated in 9. To do so, let $q > 0$ and let us denote $f_q$ the probability density function associated with the multivariate Gaussian distribution $\rho_q \sim \mathcal{N}\left(0, \frac{q}{4\varepsilon^{-1}}Id\right)$. We can rewrite the RHS of Eq. (57) as the following:*

$$\left(\frac{4}{\pi\varepsilon}\right)^{d/2}\int_{u \in \mathbb{R}^d} \exp\left(-2\varepsilon^{-1}\|x - u\|_2^2\right)\exp\left(-2\varepsilon^{-1}\|x - u\|_2^2\right)du$$

$$= \left(\frac{4}{\pi\varepsilon}\right)^{d/2}\int_{u \in \mathbb{R}^d} \exp\left(-2\varepsilon^{-1}\|x - u\|_2^2\right)\exp\left(-2\varepsilon^{-1}\|x - u\|_2^2\right)\frac{f_q(u)}{f_q(u)}d(u)$$

$$= \left(\frac{4}{\pi\varepsilon}\right)^{d/2}\int_{u \in \mathbb{R}^d} \exp\left(-2\varepsilon^{-1}\|x - u\|_2^2\right)\exp\left(-2\varepsilon^{-1}\|x - u\|_2^2\right)\left[\left(2\pi\frac{q}{4\varepsilon^{-1}}\right)^{d/2} e^{\frac{2\varepsilon^{-1}\|u\|_2^2}{q}}\right]d\rho_q(u)$$

$$= (2q)^{d/2}\int_{u \in \mathbb{R}^d} \exp\left(-2\varepsilon^{-1}\|x - u\|_2^2\right)\exp\left(-2\varepsilon^{-1}\|x - u\|_2^2\right)e^{\frac{2\varepsilon^{-1}\|u\|_2^2}{q}}d\rho_q(u)$$

*Therefore for each $q > 0$, we obtain a feature map of $k$ in $L^2(d\rho_q)$ which is defined as:*

$$\varphi(x, u) = (2q)^{d/4}\exp\left(-2\varepsilon^{-1}\|x - u\|_2^2\right)e^{\frac{\varepsilon^{-1}\|u\|_2^2}{q}}.$$

*Moreover thanks to Eq. (56) we have also:*

$$\varphi(x, u)\varphi(y, u) = (2q)^{d/2}\exp\left(-2\varepsilon^{-1}\|x - u\|_2^2\right)\exp\left(-2\varepsilon^{-1}\|y - u\|_2^2\right)e^{\frac{2\varepsilon^{-1}\|u\|_2^2}{q}}$$

$$= (2q)^{d/2}\exp\left(-\varepsilon^{-1}\|x - y\|_2^2\right)\exp\left(-4\varepsilon^{-1}\left\|u - \left(\frac{x + y}{2}\right)\right\|_2^2\right)e^{\frac{2\varepsilon^{-1}\|u\|_2^2}{q}}$$

*Therefore we have:*

$$\frac{\varphi(x,u)\varphi(y,u)}{k(x,y)} = (2q)^{d/2} \exp\left(-4\varepsilon^{-1}\left\|u - \left(\frac{x+y}{2}\right)\right\|_2^2\right) e^{\frac{2\varepsilon^{-1}\|u\|_2^2}{q}}$$

$$= (2q)^{d/2} \exp\left(-4\varepsilon^{-1}\left(1 - \frac{1}{2q}\right)\left\|u - \left(1 - \frac{1}{2q}\right)\left(\frac{x+y}{2}\right)\right\|_2^2\right)$$

$$\exp\left(\frac{4\varepsilon^{-1}}{2q-1}\left\|\left(\frac{x+y}{2}\right)\right\|_2^2\right)$$

*Finally by choosing*

$$q = \frac{\varepsilon^{-1}R^2}{2dW\left(\frac{\varepsilon^{-1}R^2}{d}\right)}$$

*where $W$ is the positive real branch of the Lambert function, we obtain that for any $x, y \in \mathcal{B}(0, R)$:*

$$0 \leq \frac{\varphi(x,u)\varphi(y,u)}{k(x,y)} \leq 2 \times (2q)^{d/2} \tag{58}$$

*Moreover we have:*

$$\varphi(x,u) = (2q)^{d/4} \exp\left(-2\varepsilon^{-1}\|x-u\|_2^2\right) e^{\frac{\varepsilon^{-1}\|u\|_2^2}{q}}$$

*Therefore $\varphi$ is differentiable with respect to $x$ and we have:*

$$\|\nabla_x\varphi\|_2^2 = 4\varepsilon^{-2}\|x-u\|_2^2 \varphi(x,u)^2 \tag{59}$$

$$\leq 4\varepsilon^{-2}\psi \sup_{x \in \mathcal{X}} k(x,x)\|x-u\|_2^2 \tag{60}$$

*where $\psi = 2 \times (2q)^{d/2}$. But by definition of the kernel we have $\sup_{x \in \mathcal{B}(0,R)} k(x,x) = 1$ and finally we have that for all $x \in \mathcal{B}(0, R)$:*

$$\mathbf{E}(\|\nabla_x\varphi\|_2^2) \leq 4\varepsilon^{-2}\psi\left[R^2 + \frac{q}{4\varepsilon^{-1}}\right] \tag{61}$$

### A.5 Another example: Arc-cosine kernel

**Lemma 3.** *Let $d \geq 1$, $s \geq 0$, $\kappa > 0$ and $k_{s,\kappa}$ be the perturbed arc-cosine kernel on $\mathbb{R}^d$ defined as for all $x, y \in \mathbb{R}^d$, $k_{s,\kappa}(x,y) = k_s(x,y) + \kappa$. Let also $\sigma > 1$, $\rho = \mathcal{N}\left(0, \sigma^2 Id\right)$ and let us define for all $x, u \in \mathbb{R}^d$ the following map:*

$$\varphi(x,u) = \left(\sigma^{d/2}\sqrt{2}\max(0, u^T x)^s \exp\left(-\frac{\|u\|^2}{4}\left[1 - \frac{1}{\sigma^2}\right]\right), \sqrt{\kappa}\right)^T$$

*Then for any $x, y \in \mathbb{R}^d$ we have:*

$$k_{s,\kappa}(x,y) = \int_{u \in \mathbb{R}^d} \varphi(x,u)^T \varphi(y,u) d\rho(u)$$

*Moreover we have for all $x, y \in \mathbb{R}^d$ $k_{s,\kappa}(x,y) \geq \kappa > 0$ and for any compact $\mathcal{X} \subset \mathbb{R}^d$ we have:*

$$\sup_{u \in \mathbb{R}^d} \sup_{(x,y) \in \mathcal{X} \times \mathcal{X}} \left|\frac{\varphi(x,u)\varphi(y,u)}{k(x,y)}\right| < +\infty \quad \text{and} \quad \sup_{x \in \mathcal{X}} \mathbf{E}(\|\nabla_x\varphi\|_2^2) < +\infty$$

**Proof.** *Let $s \geq 0$. From [13], we have that:*

$$k_s(x,y) = \int_{\mathbb{R}^d} \Theta_s(u^T x)\Theta_s(u^T y)\frac{e^{-\frac{\|u\|_2^2}{2}}}{(2\pi)^{d/2}} du$$

where $\Theta_s(w) = \max(0, w)^s$. Let $\sigma > 1$ and $f_\sigma$ the probability density function associated with the distribution $\mathcal{N}(0, \sigma^2 Id)$. Therefore we have that

$$k_s(x,y) = \int_{\mathbb{R}^d} \Theta_s(u^T x)\Theta_s(u^T y) \frac{e^{-\frac{\|u\|_2^2}{2}}}{(2\pi)^{d/2}} \frac{f_\sigma(u)}{f_\sigma(u)} du \tag{62}$$

$$= \sigma^d \int_{\mathbb{R}^d} \Theta_s(u^T x)\Theta_s(u^T y) \exp\left(-\frac{\|u\|^2}{2}\left[1 - \frac{1}{\sigma^2}\right]\right) d\rho(u) \tag{63}$$

where $\rho = \mathcal{N}(0, \sigma^2 Id)$. And by defining for all $x, u \in \mathbb{R}^d$ the following map:

$$\varphi(x, u) = \left(\sigma^{d/2}\sqrt{2}\max(0, u^T x)^s \exp\left(-\frac{\|u\|^2}{4}\left[1 - \frac{1}{\sigma^2}\right]\right), \sqrt{\kappa}\right)^T$$

we obtain that any $x, y \in \mathbb{R}^d$:

$$\int_{u \in \mathbb{R}^d} \varphi(x, u)^T \varphi(y, u) d\rho(u) = \kappa + \sigma^d \int_{\mathbb{R}^d} \Theta_s(u^T x)\Theta_s(u^T y) \exp\left(-\frac{\|u\|^2}{2}\left[1 - \frac{1}{\sigma^2}\right]\right) d\rho(u)$$

$$= \kappa + k_s(x, y)$$

$$= k_{s,\kappa}(x, y)$$

Moreover from the definition of the feature map $\varphi$, it is clear that $k_{s,\kappa} \geq \kappa > 0$,

$$\sup_{u \in \mathbb{R}^d} \sup_{(x,y) \in \mathcal{X} \times \mathcal{X}} \left| \frac{\varphi(x, u)\varphi(y, u)}{k(x, y)} \right| < +\infty \quad and \quad \sup_{x \in \mathcal{X}} \mathbf{E}(\|\nabla_x \varphi\|_2^2) < +\infty.$$

## B Constructive Method: Differentiability

### B.1 Proof of Proposition 3.2

**Proof.** *Let us first introduce the following Lemma:*
**Lemma 4.** *Let $(\alpha^*, \beta^*)$ solution of (5), then we have*

$$\max_i \alpha_i^* - \min_i \alpha_i^* \leq \varepsilon R(\mathbf{K})$$

$$\max_j \beta_j^* - \min_j \beta_j^* \leq \varepsilon R(\mathbf{K})$$

*where $R(\mathbf{K}) = -\log\left(\iota \frac{\min_{i,j} \mathbf{K}_{i,j}}{\max_{i,j} \mathbf{K}_{i,j}}\right)$ with $\iota := \min_{i,j}(a_i, b_j)$.*

**Proof B.1.** *Indeed at optimality, the primal-dual relationship between optimal variables gives us that for all $i = 1, ..., n$:*

$$e^{\alpha_i^*/\varepsilon}\langle \mathbf{K}_{i,:}, e^{\beta^*/\varepsilon}\rangle = a_i \leq 1$$

*Moreover we have that*

$$\min_{i,j} \mathbf{K}_{i,j}\langle \mathbf{1}, e^{\beta^*/\varepsilon}\rangle \leq \langle \mathbf{K}_{i,:}, e^{\beta^*/\varepsilon}\rangle \leq \max_{i,j} \mathbf{K}_{i,j}\langle \mathbf{1}, e^{\beta^*/\varepsilon}\rangle$$

*Therefore we obtain that*

$$\max_i \alpha_i^* \leq \varepsilon \log\left(\frac{1}{\min_{i,j} \mathbf{K}_{i,j}\langle \mathbf{1}, e^{\beta^*/\varepsilon}\rangle}\right)$$

*and*

$$\min_i \alpha_i^* \geq \varepsilon \log\left(\frac{\iota}{\langle \mathbf{1}, e^{\beta^*/\varepsilon}\rangle \max_{i,j} \mathbf{K}_{i,j}}\right)$$

*Therefore we obtain that*

$$\max_i \alpha_i^* - \min_i \alpha_i^* \geq -\varepsilon \log\left(\iota \frac{\min_{i,j} \mathbf{K}_{i,j}}{\max_{i,j} \mathbf{K}_{i,j}}\right)$$

*An analogue proof for $\beta^*$ leads to similar result.*

*Let us now define for any* $\mathbf{K} \in (\mathbb{R}_+^*)^{n \times m}$ *with positive entries the following objective function:*

$$F(\mathbf{K}, \alpha, \beta) := \langle \alpha, a \rangle + \langle \beta, a \rangle - \varepsilon (e^{\alpha/\varepsilon})^T \mathbf{K} e^{\beta/\varepsilon}.$$

*Let us first show that*

$$G(\mathbf{K}) := \sup_{(\alpha, \beta) \in \mathbb{R}^n \times \mathbb{R}^m} F(\mathbf{K}, \alpha, \beta) \tag{64}$$

*is differentiable on* $(\mathbb{R}_+^*)^{n \times m}$. *For that purpose let us introduce for any* $\gamma_1, \gamma_2 > 0$, *the following objective function:*

$$G_{\gamma_1, \gamma_2}(\mathbf{K}) := \sup_{\substack{(\alpha, \beta) \in B_\infty^n(0, \gamma_1) \times B_\infty^m(0, \gamma_2) \\ \alpha^T e_1 = 0}} F(\mathbf{K}, \alpha, \beta)$$

*where* $B_\infty^n(0, \gamma)$ *denote the ball of radius* $\gamma$ *according to the infinite norm and* $e_1 = (1, 0, ....0)^T \in \mathbb{R}^n$. *In the following we denote by*

$$S_{\gamma_1, \gamma_2} := \left\{ (\alpha, \beta) \in B_\infty^n(0, \gamma_1) \times B_\infty^m(0, \gamma_2) \ : \ \alpha^T e_1 = 0 \right\}.$$

*Let us now introduce the following Lemma:*

**Lemma 5.** *Let* $\varepsilon > 0$, $(a, b) \in \Delta_n \times \Delta_m$, $K \in (\mathbb{R}_+^*)^{n \times m}$ *with positive entries. Then*

$$\max_{\alpha \in \mathbb{R}^n, \beta \in \mathbb{R}^m} a^T \alpha + b^T \beta - \varepsilon (e^{\alpha/\varepsilon})^T \mathbf{K} e^{\beta/\varepsilon}$$

*admits a unique solution* $(\alpha^*, \beta^*)$ *such that* $\alpha^T e_1 = 0$, $\|\alpha^*\|_\infty \leq \varepsilon R_1(\mathbf{K})$, *and* , $\|\beta^*\|_\infty \leq \varepsilon[R_1(\mathbf{K}) + R_2(\mathbf{K})]$ *where* $R_1(\mathbf{K}) = -\log\left( \iota \frac{\min_{i,j} K_{i,j}}{\max_{i,j} K_{i,j}} \right)$, $R_2(\mathbf{K}) = \log\left( n \frac{\max_{i,j} K_{i,j}}{\iota} \right)$ *and* $\iota := \min_{i,j}(a_i, b_j)$.

**Proof B.2.** *In fact the existence and uncity up to a scalar transformation is a well known result. See for example [16]. Therefore there is a unique solution* $(\alpha^0, \beta^0)$ *such that* $(\alpha^0)^T e_1 = 0$. *Moreover thanks to Lemma 4, we have that for any* $(\alpha^*, \beta^*)$ *optimal solution that*

$$\max_i \alpha_i^* - \min_i \alpha_i^* \leq \varepsilon R(\mathbf{K}) \tag{65}$$

$$\max_j \beta_j^* - \min_j \beta_j^* \leq \varepsilon R(\mathbf{K}) \tag{66}$$

*Therefore we have* $\|\alpha^0\|_\infty \leq \max_i \alpha_i^0 - \min_i \alpha_i^0 \leq \varepsilon R(\mathbf{K})$. *Moreover, the first order optimality conditions for the dual variables* $(\alpha, \beta)$ *implies that for all* $j = 1, .., m$

$$\beta_j^0 = -\varepsilon \log \left( \sum_{i=1}^n \frac{\mathbf{K}_{i,j}}{b_j} \exp\left( \frac{\alpha_i^0}{\varepsilon} \right) \right)$$

*Therefore we have that:*

$$\|\beta^0\|_\infty \leq \|\alpha^0\|_\infty + \varepsilon \log \left( n \frac{\max_{i,j} \mathbf{K}_{i,j}}{\iota} \right)$$

*and the result follows.*

*Let* $\mathbf{K}_0 \in (\mathbb{R}_+^*)^{n \times m}$, *and let us denote* $M_0 = \max_{i,j} \mathbf{K}_0[i, j]$, $m_0 = \min_{i,j} \mathbf{K}_0[i, j]$ *and*

$$A_\omega := \left\{ \mathbf{K} \in (\mathbb{R}_+^*)^{n \times m} \quad such \ that \quad \|\mathbf{K} - \mathbf{K}_0\|_\infty < \omega \right\}$$

*By considering* $\omega_0 = \frac{m_0}{2}$, *we obtain that for any* $K \in A_{\omega_0}$,

$$R_1(\mathbf{K}) \leq \log \left( \frac{1}{\iota} \frac{2M_0 + m_0}{m_0} \right)$$

$$R_2(\mathbf{K}) \leq \log \left( n \frac{2M_0 + m_0}{2\iota} \right)$$

*Therefore by denoting*

$$\gamma_1^0 = \varepsilon \log\left(\frac{1}{\iota}\frac{2M_0 + m_0}{m_0}\right)$$

$$\gamma_2^0 = \varepsilon \left[\log\left(\frac{1}{\iota}\frac{2M_0 + m_0}{m_0}\right) + \log\left(n\frac{2M_0 + m_0}{2\iota}\right)\right]$$

*Therefore, from Lemma 5, we have that for all* $\mathbf{K} \in A_{\omega_0}$ *there exists a unique optimal solution* $(\alpha, \beta) \in B_\infty^n(0, \gamma_1^0) \times B_\infty^m(0, \gamma_2^0)$ *satisfying* $\alpha^T e_1 = 0$. *Therefore we have first that for all* $K \in A_{\omega_0}$

$$G_{\gamma_1^0, \gamma_2^0}(\mathbf{K}) = G(\mathbf{K}) \tag{67}$$

*and moreover for all* $\mathbf{K} \in A_{\omega_0}$, *the following set*

$$Z_{\mathbf{K}} := \left\{(\alpha, \beta) \in S_{\gamma_1^0, \gamma_2^0} \quad such \ that \quad F(\mathbf{K}, \alpha, \beta) = \sup_{(\alpha, \beta) \in S_{\gamma_1^0, \gamma_2^0}} F(\mathbf{K}, \alpha, \beta)\right\}$$

*is a singleton. Let us now consider the restriction of* $F$ *on* $A_{\omega_0} \times S_{\gamma_1^0, \gamma_2^0}$ *denoted* $F_0$. *It is clear from their definition that* $A_{\omega_0}$ *is an open convex set, and* $S_{\gamma_1^0, \gamma_2^0}$ *is compact. Moreover* $F_0$ *is clearly continuous, and for any* $(\alpha, \beta) \in S_{\gamma_1^0, \gamma_2^0}$, $F_0(\cdot, \alpha, \beta)$ *is convex. Moreover for any* $\mathbf{K} \in A_{\omega_0}$ *the set* $Z_{\mathbf{K}}$ *is a singleton, therefore from Danskin theorem [8], we deduce that* $G_{\gamma_1^0, \gamma_2^0}$ *is convex and differentiable on* $A_{\omega_0}$ *and we have for all* $K \in A_{\omega_0}$

$$\nabla G_{\gamma_1^0, \gamma_2^0}(\mathbf{K}) = -\varepsilon e^{\alpha^*/\varepsilon} (e^{\beta^*/\varepsilon})^T \tag{68}$$

*where* $(\alpha^*, \beta^*) \in Z_K$. *Note that any solutions of Eq.(64) can be used to evaluated* $\nabla G_{\gamma_1^0, \gamma_2^0}(\mathbf{K})$. *Moreover thanks to Eq.(67), we deduce also that* $G$ *is also differentiable on* $A_{\omega_0}$. *Finally the reasoning hold for any* $\mathbf{K}_0 \in (\mathbb{R}_+^*)^{n \times m}$, *therefore* $G$ *is differentiable and we have:*

$$\nabla G(\mathbf{K}) = -\varepsilon e^{\alpha^*/\varepsilon} (e^{\beta^*/\varepsilon})^T \tag{69}$$

## C   Illustrations and Experiments

In Figure 5, we show the time-accuracy tradeoff in the high dimensional setting. Here the samples are taken from the higgs dataset[1] [7] where the sample lives in $\mathbb{R}^{28}$. This dataset contains two class of signals: a signal process which produces Higgs bosons and a background process which does not. We take randomly 5000 samples from each of these two distributions.

In Figure 6, we consider a discretization of the positive sphere using $50^2 = 2,500$ points and generate three simple histograms of blurred pixels located in the three corners of the simplex.

Figure 5: In this experiment, we take randomly 10000 samples from the two distributions of the higgs dataset and we plot the deviation from ground truth for different regularizations. We compare the results obtained for our proposed method (**RF**) with the one proposed in [3] (**Nys**) and with the Sinkhorn algorithm (**Sin**) proposed in [16]. The cost function considered here is the square Euclidean metric and the feature map used is that presented in Lemma 1. The number of random features (or rank) chosen varies from 100 to 2000. We repeat for each problem 10 times the experiment. Note that curves in the plot start at different points corresponding to the time required for initialization. *Right, middle right*: when the regularization is sufficiently large both **Nys** and **RF** methods obtain very high accuracy with order of magnitude faster than **Sin**. *Middle left*: both methods manage to obtain high accuracy of the ROT with order of magnitude faster than **Sin**. Note that **Nys** performs better in this setting than our proposed method. *Left*: both methods fail to obtain a good approximation of the ROT.

Figure 6: Using a discretization of the positive sphere with $50^2 = 2,500$ points we generate three simple histograms (a,b,c) located in the three corners of the simplex. (d) Wasserstein barycenter with a cost $c(x, y) = -\log(x^T y)$ using the method by [9]. (e) Soft-max with temperature 1000 of that barycenter (strongly increasing the relative influence of peaks) reveals that mass is concentrated in areas that would make sense from the more usual $c(x, y) = \arccos x^T y$ distance on the sphere. The kernel corresponding to that cost, here the simple outer product of a matrix $X$ of dimsension $3 \times 2500$.