[Reviews · NeurIPS 2020]

Review 1

Summary and Contributions: This paper proposes the combination of Sinkhorn's algorithm with cost functions of the form -log <\phi(x), \phi(y)> where \phi is an embedding into nonnegative vectors. The motivation is that when the embedding vectors \phi(x) live in R^r and r is small, the Sinkhorn iteration can be computed very quickly (b/c the relevant matrix is rank r). The paper argues that some natural cost functions (e.g. square euclidean distances) can be approximated by such \phi, which lets us estimate the OT cost more quickly. A related idea from previous work (e.g. [3]) was to use a low-rank approximation as in the Nystrom method; an advantage of the approach proposed in this paper is that the needed nonnegativity property is guaranteed by construction.

Strengths: The authors propose a new way to estimate regularized OT distances which seems reasonably practical and prove that in principle it can recover the regularized OT distance under natural cost functions, note that it has some interesting properties (e.g. differentiability), and give some experiments showing the application of their method in practice.

Weaknesses: The actual quantitative bounds for approximating a cost function using nonnegative features are difficult to parse and interpret, and it's unclear how good they are in practice. For example, the bounds in Lemma 1 depend exponentially on dimension, what should we take from this?

Correctness: As far as I saw the theorems are correct and the experiments were conducted properly.

Clarity: Mostly yes. One complaint: in the experiment of Figure 1, which is supposed to compare this method with nyquist, the setup is not clear enough. "Two normal distributions" - what is the difference between the two normal distributions, what dimension are they in etc? What is the true cost function and what feature maps are being used to approximate it? Without this information it's hard to say what this Figure means.

Relation to Prior Work: Yes, it seemed to me that other relevant methods like nyquist, other variants of sinkhorn etc. have been discussed appropriately.

Reproducibility: Yes

Additional Feedback: --- post-rebuttal: thanks for answering the question about the experiment. overall my opinion is unchanged


Review 2

Summary and Contributions: In this paper, the authors propose to use random positive feature maps to calculate the kernel matrix in Sinkhorn algorithm. By using $r$ random features, the per-iteration cost of Sinkhorn iterations is reduced from $O(n^2)$ to $O(nr)$. The authors show that for kernels with forms (9), the approximate errors of the kernel matrix and calculated Sinkhorn distance can be bounded in a desirable accuracy with proper choice of $r$ under certain regularity condition of kernel function. They also show that the commonly used Arc-cosine kernels and Gaussian kernels satisfy the regularity condition and provide a constructive approach to designing positive features.

Strengths: This paper provide a practical way to reduce the per-iteration computation cost of the Sinkhorn method. The theoretical analysis is sound and comprehensive. As Wasserstein distance based methods have been widely used in machine learning, this work would have a wide audience in the NeuraIPS community.

Weaknesses: My major concern are listed as follows. Random feature maps have been widely adopted in approximating the kernel matrix, since any kernel function can be approximate by eigenfunctions according to the Mercer’s theorem. Here, the authors restrict the feature maps to positive as $K = exp(-C/\epsilon) > 0$ and require the kernel function satisfies certain regularity conditions. Thus, this method is not suitable for all cost function $c$. $r$ is not guaranteed to be small though its dependency on $n$ is $O(\log(n))$. For example, for the Gaussian kernels, $\phi$ and $V$ is $O(2^d)$, which indicates $r = \Omega(\phi^2)$ is also $O(2^d)$. Thus, this method would need a large $r$ even for moderate $d$. The empirical studies are somehow weak as the authors only compares different methods on calculating the Sinkhorn distance between two normal distributions(Exp1). In the second experiments, the authors shows that this method can learn the adversarial kernel function. However, the results in Table 1 is a little confused: The quantity is $k_\theta(f_\gamma(x),f_\gamma(y))$ or $c_\theta(f_\gamma(x),f_\gamma(y))$? As $k = exp(-c/\epsilon)$, k should be smaller than 1. If it is $c$, why the distance between z and z is much bigger than x and z? =======update after author rebuttal======== My major concern is that $r$, the number of features, exponentially depends on the the problem dimension $d$ for the most important Gaussian case. The authors acknowledge this drawback. Also, I think approximating kernel with random features according to the Mercer’s theorem is a very standard technique. The novelty of this paper is hence limited.

Correctness: The claims are correct and the empirical methodology is correct.

Clarity: This paper is well organized and well written.

Relation to Prior Work: The relation to previous contributions is clearly discussed

Reproducibility: Yes

Additional Feedback: What is performance of Sinkhorn algorithm with other direct rank r approximation of matrix $K$ such as rank-r SVD?


Review 3

Summary and Contributions: The paper proposes a method for approximating the optimal transport and Sinkhorn divergence costs by exploiting the feature representation of the kernel matrix. This approach leads to faster faster Sinkhorn iterations and is still fully differentiable.

Strengths: The approach is interesting theoretically, and well validated empirically. There exist previous approaches that try to do the same thing, but they are either slower or non-differentiable or both. The paper is clearly written, and the contribution is easy to understand, the proofs easy to follow.

Weaknesses: The section on generative adversarial networks needs more detail. As it is now, it reads like a last minute addition. The authors mention an alternative approach via batching the input and training a normal W-GAN but there is no comparison with this approach either quantitative or qualitative. It would also be great to see how the approach compares to Sinkhorn/Nystrom on different manifolds and metrics. Remark 1 mentions transport on a sphere for example.

Correctness: The statements present in the paper are correct to my understanding.

Clarity: The paper is clearly written and understandable. The theorems and proofs are easy to follow.

Relation to Prior Work: All relevant prior work is cited.

Reproducibility: Yes

Additional Feedback: Post rebuttal: ----------------- My opinion of the paper has not changed. The authors promise more detail and experiments. If these are to be provided, I am strongly in favor of accepting this paper.

[Author Response · NeurIPS 2020]

¹ **We thank the reviewers for their thorough reading of our work.** We will use their remarks to improve our draft.

² Reviewer #1: ■ *The actual quantitative bounds for approximating a cost function using nonnegative features are* ³ *difficult to parse and interpret.* We agree this is a bit dry. Our result in Prop. 3.1 gives a uniform control of the ⁴ *ratio* between the kernel and its approximation. This can be compared to the classical result in [46] to control the ⁵ *difference* between them. In the Sinkhorn setting, controlling the ratio is more important than controlling the difference ⁶ (L.161), because it ensures that *the approximation retains the same positive sign.* We will discuss this further. ■ *For* ⁷ *example, the bounds in Lemma 1 depend exponentially on dimension, what should we take from this?* When studying ⁸ the Gaussian kernel in Lemma 1, both constants $\psi$ and $V$ that are upper bounds. While we can envision ways to obtain ⁹ tighter, dimension free upper bounds using additional assumptions, we do indeed pay a price in dimension here when ¹⁰ tackling the Euclidean distance with full generality. ■ *In the experiment of Figure 1, which is supposed to compare this* ¹¹ *method with nyquist, the setup is not clear enough.* Indeed, this must be clarified. These two normal distributions are in ¹² $\mathbb{R}^2$. One of them has mean $(1,1)^T$ and identity covariance matrix $I_2$. The other has 0 mean and covariance $0.1 \times I_2$. ¹³ The cost function is the square Euclidean metric and the feature map is that presented in Lemma 1.

¹⁴ Reviewer #2: ■ *Thus, this method is not suitable for all cost function c.* Indeed, while we do stress the ability of our ¹⁵ low-rank kernels to approximate known kernels induced by classical distances, some distances will prove elusive. For ¹⁶ instance, if the support of both measures coincide, if the distance of interest is *not* Hilbertian, the resulting square kernel ¹⁷ matrix $K$ is *not* even positive definite, and certainly not approximated with low rank factors. Obtaining positive factors ¹⁸ whose product can approximate $K$ is therefore totally hopeless. This is why we also stress a *constructive* approach ¹⁹ in our work, one that stresses the importance of considering families of kernels for which Sinkhorn has linear time ²⁰ complexity. ■ *r is not guaranteed to be small though its dependency on n is $O(\log n)$.* The fact that the dependency of ²¹ $r$ on $n$ is $O(\log n)$ does not really affect the required number of random features as it depends logarithmically on the ²² number of samples. However the constant $\psi$ has a more important role. In the specific case of the Gaussian kernel, it is ²³ true that the constant $\psi$ exhibited in the paper depends exponentially in the dimension but recall that this is an upper ²⁴ bound and it may be loose. Moreover the dependence in the dimension may be removed in some cases: for example ²⁵ if the data lies in a specific region of the space, then one may drop the dependence in the dimension in the bound by ²⁶ sampling the features on this low dimensional manifold. ■ *The empirical studies are somehow weak as the authors* ²⁷ *only compares different methods on calculating the Sinkhorn distance between two normal distributions(Exp1).* We ²⁸ used this setup for its simplicity, but we agree that another experiment would be welcome. We will prepare it for our ²⁹ next version. Moreover we will also add an experiment in a high-dimensional setting, to see how the proposed method ³⁰ scales with respect to the dimension in the specific case of the square Euclidean cost. ■ *As $k = exp(-c/\epsilon)$, k should* ³¹ *be smaller than 1.* Because we flip things upside down and start directly with a kernel (not a cost), and the kernel itself ³² is learned in an adversarial manner as the product of positive factors, the kernel can take arbitrarily large values. If ³³ they were recast (somewhat artificially) as cost functions, these values would indeed correspond to a negative cost. We ³⁴ also take this opportunity to correct a typo in Table 2: Image $x$ / Noise $z$ should be flipped both in lines and columns ³⁵ descriptions. ■ *What is performance of Sinkhorn algorithm with other direct rank r approximation of matrix K such* ³⁶ *as rank-r SVD?* As we mention in L.48-51 (and is discussed in [3]) using arbitrary low-rank factorizations for $K$ fails if ³⁷ no proper care is taken to ensure that each factor has positive entries. This is not guaranteed for SVD, neither in theory ³⁸ nor have we observed it in practice. This is needed because the Sinkhorn iterations in Alg.1 use elementwise *divisions* ³⁹ of kernel products. We will clarify. Additionally, r-SVD would not allow a linear regime.

⁴⁰ Reviewer #4: ■ *The section on generative adversarial networks needs more detail.* We will add more explanations ⁴¹ on how to implement our method to train a W-GAN in the final version. Indeed here we embed the image space into ⁴² the feature map space thanks to two operations. The first one consists in taking an image and embedding it into a ⁴³ latent space thanks to the mapping $f_\gamma$ and the second one is an embedding of this latent space into the feature space ⁴⁴ thanks to the feature map $\varphi_\theta$. Here the mapping considered is a parametrized version of the feature map defined in ⁴⁵ Lemma 1 obtained from the Gaussian kernel. ■ *The authors mention an alternative approach via batching the input* ⁴⁶ *and training a normal W-GAN but there is no comparison with this approach either quantitative or qualitative.* We ⁴⁷ will also add a discussion to compare with other W-GANs. Indeed our method has mainly two advantages compared to ⁴⁸ the other W-GANs proposed in the literature. First, the computation of the Sinkhorn divergence is linear with respect ⁴⁹ to the number of samples which allow to largely increase the batch size when training a W-GAN and obtain a better ⁵⁰ approximation of the true Sinkhorn divergence. Second, our approach is fully differentiable and therefore we can ⁵¹ directly compute the gradient of the Sinhkorn divergence with respect the parameters of the network. In [48] the authors ⁵² do not differentiate through the Wasserstein cost to train their network. In [27] the authors do differentiate through the ⁵³ iterations of the Sinkhorn algorithm but this strategy require to keep track of the computation involved in the Sinkhorn ⁵⁴ algorithm and can be applied only for large regularizations as the number of iterations cannot be too large. ■ *It would* ⁵⁵ *also be great to see how the approach compares to Sinkhorn/Nystrom on different manifolds and metrics.* We will add ⁵⁶ another experiment to compare our proposed approach with the Nystrom method on different manifolds in the final ⁵⁷ version.

[Meta-Review · NeurIPS 2020]

This paper proposes the combination of Sinkhorn's algorithm with cost functions of the form -log <\phi(x), \phi(y)> where \phi is an embedding into nonnegative vectors. The motivation is that when the embedding vectors \phi(x) live in R^r and r is small, the Sinkhorn iteration can be computed very quickly (b/c the relevant matrix is rank r). There are some issues with practical applications given that the actual quantitative bounds for approximating a cost function using nonnegative features. For e.g., even for Gaussian kernels, the rank turns out to be exponential in the dimension so it's not clear if the methods are practical. Despite this, the reviewers found the research direction important and were convinced of the importance of the algorithm proposed in the work. I am pleased to recommend acceptance.